# Direct Alignment with Heterogeneous Preferences

**Ali Shirali**[*]
UC Berkeley

**Arash Nasr-Esfahany**[*]
MIT

**Abdullah Alomar**
MIT & Ikigai Labs

**Parsa Mirtaheri**
UC San Diego

**Rediet Abebe**[†]
ELLIS Institute, MPI for Intelligent Systems, & Tübingen AI Center

**Ariel Procaccia**[†]
Harvard University

## Abstract

Alignment with human preferences is commonly framed using a universal reward function, even though human preferences are inherently heterogeneous. We formalize this heterogeneity by introducing *user types* and examine the limits of the homogeneity assumption. We show that aligning to heterogeneous preferences with a single policy is best achieved using the average reward across user types. However, this requires additional information about annotators. We examine improvements under different information settings, focusing on direct alignment methods. We find that minimal information can yield first-order improvements, while full feedback from each user type leads to consistent learning of the optimal policy. Surprisingly, however, no sample-efficient consistent direct loss exists in this latter setting. These results reveal a fundamental tension between consistency and sample efficiency in direct policy alignment.

## 1 Introduction

Human rewards and preferences are heterogeneous [1–5]. Despite this, learning from preference data often bypasses this insight, relying on what we dub the *preference homogeneity assumption*. This tension in assumptions is readily apparent in standard human-AI alignment methods—such as reinforcement learning from human feedback (RLHF) [6–9] and direct preference optimization (DPO) [10]—which assume a single reward function captures the interests of the entire population.

We study a heterogeneous population in which individuals belong to distinct *user types*, each defined by its own reward function. Two main approaches address this setting [11]: *personalizing* policies for each user type [12–17], or *aggregating* diverse rewards into a single objective and deploying a universal policy [18, 19]. Our focus is on the latter, which is necessary when user types are unobservable at inference time [20] or when the cost of maintaining specialized models is prohibitive. A universal policy is also preferable when personalization undermines core values—such as truth or safety—that benefit from broad consensus [21, 22].

One may notice that standard methods based on the preference homogeneity assumption implicitly aggregate rewards. However, such aggregation can be undesirable or counterintuitive, leading to unexpected behavior [23–25], as learning from heterogeneous preferences becomes *unrealizable*: A single reward cannot capture the complexity of population preferences with multiple rewards [26, 11].

In the quest for a single policy that accommodates a heterogeneous population, we show that the only aggregation of rewards across user types that yields a well-defined alignment problem is an affine aggregation, with the average reward emerging as a natural choice. However, standard methods like DPO do not maximize this user-weighted average reward. Building on insights by Siththaranjan et al.

---

[*]AS and AN contributed equally. AS conducted this work while visiting Harvard.
[†]Equal advising

[20], we show that DPO implicitly maximizes Borda count, which comes with unexpected drawbacks, e.g., the optimal solution depends on how alternative responses are sampled, even for infinite data.

We observe that learning the average reward over user types—or equivalently, a policy that maximizes it—from anonymous data is impossible. Focusing on *direct alignment methods* [10, 27, 28] which avoid explicit reward modeling, we study the benefits of using annotator data for a range of information settings. We show that improving DPO with a first-order correction to its objective is possible with minimal annotator information. Specifically, we design an approximate direct alignment method when each preference data point is paired with another one labeled by the same user.

On the other hand, we find that there are limits to what is possible even with significant annotator information. In particular, we propose a consistent loss function for direct alignment when we have feedback on each data point from every user type. But this loss is sample-inefficient, using only data where all annotators agreed. Surprisingly, we prove that no consistent loss uses the rest of the data.

In sum, the homogeneity assumption leads to undesirable outcomes when aligning a single AI agent to diverse preferences. Our analysis shows that there is a limited class of reward aggregation that yields a well-defined alignment objective, with average reward over user types emerging as the natural choice. This requires some annotator data, though small amounts of data can yield significant improvements. Our findings, however, uncover a fundamental tension between consistency and sample efficiency in direct alignment: To achieve both, we must forgo the benefits of direct optimization and instead train individualized reward models, which inevitably incurs significant training and storage costs.

## 2 Preliminaries

In the *alignment problem*, we consider a setting where a *reward* function $r^*$ evaluates a response $\boldsymbol{y}$ to a query $\boldsymbol{x}$ by assigning a score $r^*([\boldsymbol{x}, \boldsymbol{y}])$. The goal is to design a *policy* that selects high-reward responses for each query. Formally, a policy $\pi$ defines a probability distribution over responses: given a query $\boldsymbol{x}$, it selects $\boldsymbol{y}$ with probability $\pi(\boldsymbol{y} \mid \boldsymbol{x})$. Typically, we begin with a reference policy $\pi_{\text{ref}}$, which acts as a prior over $\boldsymbol{y}$ [29], and for each $\boldsymbol{x}$, aim to maximize

$$\mathbb{E}_{\boldsymbol{y} \sim \pi(\cdot \mid \boldsymbol{x})}\big[r^*([\boldsymbol{x}, \boldsymbol{y}])\big] - \beta D_{\text{KL}}\big(\pi(\cdot \mid \boldsymbol{x}); \pi_{\text{ref}}(\cdot \mid \boldsymbol{x})\big), \tag{1}$$

where $D_{\text{KL}}$ is the KL-divergence. We denote the optimal policy by $\pi^*$. In practice, $\pi_{\text{ref}}$ is often a pretrained (language) model, and the regularization parameter $\beta$ controls deviation from it. Eq. (1) often includes $\mathbb{E}_{\boldsymbol{x}}$, which is important in practice but does not affect $\pi^*$ in theory.

When $r^*$ is known, we can directly apply RL to maximize Eq. (1). In practice, however, $r^*$ is often unknown and must be inferred from human feedback before applying RL, a process known as RLHF. Although widely used, RLHF can be difficult to tune due to the complexities of RL. An alternative gaining traction is *direct alignment with preferences* [30, 10, 27], which avoids explicit reward modeling and instead trains the policy directly from human feedback.

**Preference Model.** Both direct alignment and RLHF rely on a model of human preference to relate reward values with observed preference data. Consider the case where responses $\boldsymbol{y}_1$ and $\boldsymbol{y}_2$ are generated for a given query $\boldsymbol{x}$. We express the probability that $\boldsymbol{y}_2$ is preferred to $\boldsymbol{y}_1$ as

$$\Pr(\boldsymbol{y}_2 \succ \boldsymbol{y}_1 \mid \boldsymbol{x}; r^*) = \sigma\big(r^*([\boldsymbol{x}, \boldsymbol{y}_2]) - r^*([\boldsymbol{x}, \boldsymbol{y}_1])\big), \tag{2}$$

where $\sigma$ is a non-decreasing function bounded between 0 and 1 [31–33]. A widely-used choice for $\sigma$ is the sigmoid function corresponding to the well-known Bradley-Terry (BT) model [34]. Related work have explored smooth [35] or inconsistent preferences [36, 37], which are beyond our scope.

**Direct Preference Optimization.** Among direct alignment methods, DPO has become the de facto standard. By exploiting a closed-form solution for maximizing Eq. (1), it directly relates any reward to its optimal policy. Thereby, instead of estimating the reward explicitly, DPO trains a policy whose *induced reward* best accounts for the observed preferences. Below, we derive this connection.

First, maximizing Eq. (1) has a well-known solution [38]. The optimal policy $\pi^*$ takes the form

$$\pi^*(\boldsymbol{y} \mid \boldsymbol{x}) = \frac{1}{Z(\boldsymbol{x})} \pi_{\text{ref}}(\boldsymbol{y} \mid \boldsymbol{x}) \cdot \exp\left(\frac{1}{\beta} r^*([\boldsymbol{x}, \boldsymbol{y}])\right). \tag{3}$$

Here, $Z(\boldsymbol{x}) = \sum_{\boldsymbol{y}'} \pi_{\text{ref}}(\boldsymbol{y}' \mid \boldsymbol{x}) \cdot \exp(\frac{1}{\beta} r^*([\boldsymbol{x}, \boldsymbol{y}']))$ is the partition function. Eq. (3) establishes a direct relationship between policy ratios and reward differences as

$$r^*(\boldsymbol{y}_2) - r^*(\boldsymbol{y}_1) = \beta \log \frac{\pi^*(\boldsymbol{y}_2)}{\pi_{\text{ref}}(\boldsymbol{y}_2)} - \beta \log \frac{\pi^*(\boldsymbol{y}_1)}{\pi_{\text{ref}}(\boldsymbol{y}_1)} . \tag{4}$$

Henceforth, we omit $\boldsymbol{x}$ when clear from context. This shows that the reward difference between two responses is fully captured by the difference in their policy ratios, and motivates the definition of the *induced reward* of a policy $\pi$ as $\beta \log \frac{\pi(\boldsymbol{y})}{\pi_{\text{ref}}(\boldsymbol{y})}$ or the reward function for which $\pi$ is the optimal policy.

The difference in rewards of $\boldsymbol{y}_1$ and $\boldsymbol{y}_2$ is sufficient to express the likelihood of $\boldsymbol{y}_2 \succ \boldsymbol{y}_1$ in Eq. (2). Using Eq. (4), we can therefore write the likelihood as a function of $\pi^*$:

$$\Pr(\boldsymbol{y}_2 \succ \boldsymbol{y}_1 \mid \pi^*) = \sigma\Big(\beta \log \frac{\pi^*(\boldsymbol{y}_2)}{\pi_{\text{ref}}(\boldsymbol{y}_2)} - \beta \log \frac{\pi^*(\boldsymbol{y}_1)}{\pi_{\text{ref}}(\boldsymbol{y}_1)}\Big). \tag{5}$$

For any policy $\pi$, we similarly define $\Pr(\boldsymbol{y}_2 \succ \boldsymbol{y}_1 \mid \pi)$. To estimate $\pi^*$, we can then apply maximum likelihood estimation: Given a dataset $\mathcal{D}$ of query-response pairs $(\boldsymbol{x}, \boldsymbol{y}_l, \boldsymbol{y}_w)$ with $\boldsymbol{y}_w \succ \boldsymbol{y}_l$, DPO maximizes the following log-likelihood (or equivalently, minimizes the cross-entropy loss):

$$\sum_{(\boldsymbol{x}, \boldsymbol{y}_l, \boldsymbol{y}_w) \in \mathcal{D}} \log \Pr(\boldsymbol{y}_w \succ \boldsymbol{y}_l \mid \boldsymbol{x}; \pi), \tag{6}$$

## 3  Problem Formulation

The alignment problem is traditionally framed under a preference homogeneity assumption, where a single reward is presumed to capture all individual interests. In practice, people's preferences can differ significantly. To better capture real-world settings, we formalize preference heterogeneity by allowing reward functions to vary across *user types* as we detail below.

**Heterogeneous Preference Model.** The influential study of "individual choice behavior" by Luce [32] and other foundational works on human decision-making in mathematical psychology such as Shepard [39] focus on *individual* preference models. Luce [32] uses an axiomatic framework to establish the existence of a value function—akin to a reward—for *each individual*, which, once normalized, explains their choice probabilities ($\exp(r^*)$ in the BT model is one such value function).

In practice, individual identities are often unobserved. As a result, standard preference modeling approaches assume a single reward function for the entire population. This homogeneity assumption renders preference learning *unrealizable*: Even if a family of models can represent each individual's preferences, it may fail to capture aggregate population behavior. For instance, a mixture of BT models cannot be represented by a single BT (we prove this in Prop. F.1 for completeness).

To fully account for heterogeneity, we need to define individual rewards. However, learning at scale with this level of granularity is impractical, especially when working with finite data. Hence, we group individuals into multiple *user types*, denoted by $\mathcal{U}$, where individuals with the same type have similar rewards, but this need not hold across types. Unless otherwise stated, we assume $|\mathcal{U}| > 1$.

For a user of type $u \in \mathcal{U}$, we denote their reward function by $r^*(\cdot; u)$, which assigns a score $r^*(\boldsymbol{y}; u)$ to each response $\boldsymbol{y}$. We model the preferences of users of type $u$ as

$$\Pr(\boldsymbol{y}_2 \succ \boldsymbol{y}_1 \mid r^*, u) = \sigma\big(r^*(\boldsymbol{y}_2; u) - r^*(\boldsymbol{y}_1; u)\big). \tag{7}$$

The law of total expectation then implies that the population-level preferences follow

$$\Pr(\boldsymbol{y}_2 \succ \boldsymbol{y}_1 \mid r^*) = \mathbb{E}_u\Big[\sigma\big(r^*(\boldsymbol{y}_2; u) - r^*(\boldsymbol{y}_1; u)\big)\Big]. \tag{8}$$

**The Extended Alignment Problem.** Deriving a universal policy to serve a heterogeneous population requires aggregation of diverse rewards. As we show next, an affine combination is the only form of aggregation that guarantees a well-defined problem, i.e., a problem that yields the same optimal policy for every reward that is consistent with the distribution of preferences.

**Proposition 3.1.** *Consider a differentiable aggregation function $f : \mathbb{R}^{\mathcal{U}} \to \mathbb{R}$. Suppose that for every reward $r$ consistent with the preference distribution—meaning $\Pr(\boldsymbol{y}_2 \succ \boldsymbol{y}_1 \mid r) = \Pr(\boldsymbol{y}_2 \succ \boldsymbol{y}_1 \mid r^*)$ for all prompts and responses $\boldsymbol{y}_1, \boldsymbol{y}_2$ in the response space $\mathcal{Y}$—the function $f\big((r(\boldsymbol{y}; u))_{u \in \mathcal{U}}\big)$ induces the same ordering over $\mathcal{Y}$. If $\{(r(\boldsymbol{y}; u))_{u \in \mathcal{U}} \mid \boldsymbol{y} \in \mathcal{Y}\}$ has non-empty interior, then $f$ must be affine.*

See proof on page 25. This result rules out many commonly-used aggregations, such as Max-Min [19] or Nash social welfare [40]. Among all the affine combinations, the expected reward across user types emerges as a natural choice here. Any other affine combination would weigh people unequally, which requires strong justifications and is rare in practice.

To summarize, our objective is to maximize

$$\mathbb{E}_{\boldsymbol{y} \sim \pi(\cdot | \boldsymbol{x})}\big[\mathbb{E}_u\big[r^*([\boldsymbol{x}, \boldsymbol{y}]; u)\big]\big] - \beta D_{\mathrm{KL}}\big(\pi(\cdot \mid \boldsymbol{x}); \pi_{\mathrm{ref}}(\cdot \mid \boldsymbol{x})\big), \tag{9}$$

for every prompt $\boldsymbol{x}$. With this extended framework in mind, we next discuss why standard approaches like RLHF or DPO do not necessarily yield the optimal policy.

## 4 Implications of the Preference Homogeneity Assumption

When preferences are heterogeneous, standard RLHF or DPO cannot yield the optimal policy $\pi^*$ that maximizes Eq. (9). If they could, the (user-weighted) expected reward would be learnable as the induced reward of $\pi^*$. However, as we show in Prop. 5.1—and as noted in prior work in specific cases [20, 41]—learning the expected reward from anonymous preferences is impossible.

To explain DPO's failure in finding $\pi^*$, we extend its derivation to the heterogeneous setting in Sec. 4.1, laying the groundwork to account for heterogeneity in DPO later on. In Sec. 4.2, we show DPO's policy aligns with Borda count and, in Sec. 4.3, highlight its limitations. While our analysis focuses on DPO, similar insights extend to RLHF by substituting the policy with its induced reward.

### 4.1 Objective is Not the Expected Reward

We follow DPO's derivation from Sec. 2 but with heterogeneous preferences. We show that the closed-form connection between $\pi^*$ and $r^*$ is no longer sufficient to express the likelihood function. Starting from Eq. (9), the optimal policy is

$$\pi^*(\boldsymbol{y}) = \frac{1}{Z(\boldsymbol{x})} \pi_{\mathrm{ref}}(\boldsymbol{y}) \cdot \exp\left(\frac{1}{\beta} \mathbb{E}_u\big[r^*(\boldsymbol{y}; u)\big]\right). \tag{10}$$

Define $\Delta r^*(\boldsymbol{y}_1, \boldsymbol{y}_2; u) := r^*(\boldsymbol{y}_2; u) - r^*(\boldsymbol{y}_1; u)$. Using the above solution, the policy ratios of $\pi^*$ relate to the expected difference in rewards as

$$\mathbb{E}_u\big[\Delta r^*(\boldsymbol{y}_1, \boldsymbol{y}_2; u)\big] = \beta \log \frac{\pi^*(\boldsymbol{y}_2)}{\pi_{\mathrm{ref}}(\boldsymbol{y}_2)} - \beta \log \frac{\pi^*(\boldsymbol{y}_1)}{\pi_{\mathrm{ref}}(\boldsymbol{y}_1)}. \tag{11}$$

In the homogeneous case, $\Delta r^*$ was sufficient to describe the likelihood of $\boldsymbol{y}_2 \succ \boldsymbol{y}_1$. However, with heterogeneous preferences, $\mathbb{E}_u[\Delta r^*]$ alone does not suffice to write the likelihood function in Eq. (8). Only under the following approximation can we express $\Pr(\boldsymbol{y}_2 \succ \boldsymbol{y}_1 \mid r^*)$ in terms of policy ratios, as in Eq. (5), and minimize DPO's loss to recover $\pi^*$:

$$\mathbb{E}_u\Big[\sigma\big(\Delta r^*(\boldsymbol{y}_1, \boldsymbol{y}_2; u)\big)\Big] \approx \sigma\Big(\mathbb{E}_u\big[\Delta r^*(\boldsymbol{y}_1, \boldsymbol{y}_2; u)\big]\Big). \tag{12}$$

### 4.2 Ordinal Consistency with Borda Count

If DPO were the answer, what would the question be? We partially answer this question by an adaptation of a result from Siththaranjan et al. [20]. First, define the Borda count as follows.

**Definition 4.1** (Normalized Borda count). For a prompt $\boldsymbol{x}$, let $\mathcal{D}(\cdot \mid \boldsymbol{x})$ denote the distribution of alternative responses sampled for $\boldsymbol{x}$. The Normalized Borda Count (NBC) of $\boldsymbol{y}$ at $\boldsymbol{x}$ is the probability that an annotator with a random type prefers $\boldsymbol{y}$ over an alternative response $\boldsymbol{y}' \sim \mathcal{D}(\cdot \mid \boldsymbol{x})$:

$$\mathrm{NBC}(\boldsymbol{y} \mid \boldsymbol{x}) := \mathbb{E}_{\boldsymbol{y}' \sim \mathcal{D}(\cdot | \boldsymbol{x})}\Big[\Pr(\boldsymbol{y} \succ \boldsymbol{y}' \mid \boldsymbol{x}; r^*)\Big]. \tag{13}$$

We next show that DPO's policy ratios are ordinally consistent with the normalized Borda count.

**Proposition 4.2.** *Suppose responses to $\boldsymbol{x}$ in the preference dataset are drawn from $\mathcal{D}(\cdot|\boldsymbol{x})$ and labeled according to BT. In the population limit, where empirical averages converge to expectations, DPO's induced reward, or equivalently $\frac{\pi_{\mathrm{DPO}}(\cdot|\boldsymbol{x})}{\pi_{\mathrm{ref}}(\cdot|\boldsymbol{x})}$, has the same ordering over responses as $\mathrm{NBC}(\cdot \mid \boldsymbol{x})$.*[3]

See proof on page 25. Prop. 4.2 also applies to the homogeneous setting. In this case, however, NBC aligns with $r^*$. It is worth mentioning that DPO is not the only method consistent with NBC; identity preference optimization (IPO) [42] uses NBC as its objective. We next highlight key differences between NBC and the user-weighted expected reward along with DPO's drawbacks in practice.

### 4.3 Practical Drawbacks

Borda count can significantly diverge from the user-weighted expected reward. This is studied under *distortion* in social choice theory [43]. Notably, NBC in Eq. (13) depends on $\mathcal{D}$. Thus, while data collection is irrelevant to defining the optimal policy, it affects $\pi_{\mathrm{DPO}}$. Next, we illustrate two key differences between $\pi^*$ and $\pi_{\mathrm{DPO}}$ through examples. Unless otherwise stated, we assume $\mathcal{D}(\cdot \mid \boldsymbol{x})$ and $\pi_{\mathrm{ref}}(\cdot \mid \boldsymbol{x})$ are uniform, and annotators follow BT. Refer to Appendix A for further drawbacks of DPO (namely, minority suppression and IIA violation).

**Mediocrity Promotion.** Consider a summarization task with $\mathcal{U} = \{A, B, C\}$, where all types are equally represented. Type $A$ strongly prefers longer summaries, type $B$ strongly prefers shorter ones, and type $C$ slightly prefers medium-length summaries:

$$r^*(\text{short}; A) = 0, \ r^*(\text{med}; A) = 1, \ r^*(\text{long}; A) = 4,$$
$$r^*(\text{short}; B) = 4, \ r^*(\text{med}; B) = 1, \ r^*(\text{long}; B) = 0,$$
$$r^*(\text{short}; C) = 0, \ r^*(\text{med}; C) = 1, \ r^*(\text{long}; C) = 0.$$

In this case, $\pi^*(\text{short}) = \pi^*(\text{long}) > \pi^*(\text{med})$, but, $\mathrm{NBC}(\text{short}) = \mathrm{NBC}(\text{long}) < \mathrm{NBC}(\text{med})$. Hence, DPO prefers medium-length summaries that are not strongly favored by any type.

**Sensitivity to Preference Dataset Distribution.** Suppose $\mathcal{U} = \{A, B\}$ and types are equally represented. Given three possible responses, type $A$ prefers $\boldsymbol{y}_1$ but type $B$ prefers $\boldsymbol{y}_2$:

$$r^*(\boldsymbol{y}_1; A) = 6, \ r^*(\boldsymbol{y}_2; A) = 1, \ r^*(\boldsymbol{y}_3; A) = 4,$$
$$r^*(\boldsymbol{y}_1; B) = 3, \ r^*(\boldsymbol{y}_2; B) = 9, \ r^*(\boldsymbol{y}_3; B) = 4.$$

One can verify that while $\mathcal{D}(\boldsymbol{y}_1) = \mathcal{D}(\boldsymbol{y}_2)$, increasing $\mathcal{D}(\boldsymbol{y}_3)$ from 0.02 to 0.04 changes $\pi_{\mathrm{DPO}}$'s preference from $\boldsymbol{y}_2$ to $\boldsymbol{y}_1$. DPO's policy is also sensitive to the preference model. Consider a variation of BT with a temperature of 2: $\sigma_2(z) := (1 + \exp(-z/2))^{-1}$. For the same users and uniform sampling of alternatives, increasing the temperature from 1 to 2 flips $\pi_{\mathrm{DPO}}$'s ranking over $\boldsymbol{y}_1$ and $\boldsymbol{y}_2$ while the preference model has no effect on $\pi^*$ as expected from the optimal policy.

We have to emphasize that the dependence of NBC, and consequently $\pi_{\mathrm{DPO}}$, on the dataset sampling distribution $\mathcal{D}$ is not due to finite-sample limitations or insufficient offline dataset support. This issue persists even with complete data coverage and in the limit of infinite data.

**Real-World Examples.** The examples above are not contrived; in real-world cases, NBC can produce rankings different from $\pi^*$ and is sensitive to dataset distribution as extensively studied under distortion of social choice rules. To show this with a real example, we use Pew Research Center surveys and analyze a question to 5101 participants: "The next time you purchase a vehicle, how likely are you to consider purchasing an electric vehicle?" (options from A: very likely to D: not at all likely). We discuss how we select this question in Appendix B. Responses come from groups of different political leanings: Republican (45%), Democratic (48%), and Neither/refused (7%).

Assuming the Luce-Shepard model [39] (see Eq. (18)), we estimate the reward for each group to calculate NBC and a user-weighted average reward. To find NBC, we use two distributions for alternatives: a uniform distribution $\mathcal{D}_U$ and a slightly altered distribution $\mathcal{D}_a$ with 0.2 total variation distance (TV) from $\mathcal{D}_U$. As shown in Fig. 1, NBC (with $\mathcal{D}_U$) ranks option C first despite its mediocrity: it is the second or third preference for the three groups (see Fig. 7). In contrast, the user-weighted average reward favors D: the first and second preference for Republicans and the no-lean

---

[3]We can view $\mathrm{NBC}(\boldsymbol{y} \mid \boldsymbol{x})$ as an aggregation of rewards at $\boldsymbol{y}$. One can verify that NBC meets the order consistency condition of Prop. 3.1. However, it uses the reward value at $\boldsymbol{y}' \neq \boldsymbol{y}$ to define the aggregated reward at $\boldsymbol{y}$ and thus does not fall under Prop. 3.1. In fact, this interdependency causes the issues we discuss Sec. 4.3.

groups, respectively. Notably, altering $\mathcal{D}_U$ to $\mathcal{D}_a$ flips NBC's top ranking, highlighting its sensitivity to dataset distribution. Similar discrepancies appear in other Pew surveys (see Appendix B).

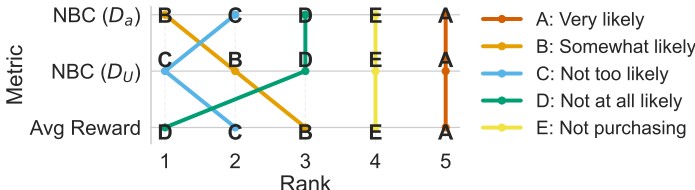

Figure 1: Ranking of different options in response to the question "The next time you purchase a vehicle, how likely are you to seriously consider purchasing an EV?" To find NBC, we use two distributions for alternatives: a uniform distribution $\mathcal{D}_U$ and a slightly altered distribution $\mathcal{D}_a$ with 0.2 TV distance from $\mathcal{D}_U$. NBC ranking is sensitive to the dataset distribution.

# 5 Approximate Direct Alignment with Minimal Annotator Information

The failure of standard alignment methods to recover the optimal policy $\pi^*$ raises a natural question: is it even possible to identify $\pi^*$? Without annotator information, the answer is no. To show this, it suffices to prove that the ranking based on the (user-weighted) expected reward is not learnable. This implies that $\pi^*$ is also not learnable since its induced reward corresponds to the expected reward by definition. We defer the formal definition of learnability to Def. F.2 in the appendix, and based on it, we prove the following impossibility result:

**Proposition 5.1.** *When the preference model $\sigma(\cdot)$ is continuous, the ranking based on the (user-weighted) expected reward is not learnable (according to Def. F.2) without annotator information.*

See proof on page 26. Siththaranjan et al. [20] (Theorem. 3.4) presented a version of Prop. 5.1 for two alternatives and two types when $\sigma(\Delta r) = \mathbb{1}\{\Delta r > 0\}$. Procaccia et al. [41] (Theorem. 2.2) generalized this to BT. Prop. 5.1 presents a fresh perspective by generalizing the impossibility to any continuous preference model and presenting multiple proof strategies, including one that draws on a robust version of Arrow's theorem [44].

To circumvent the impossibility in Prop. 5.1, we must either relax the requirement of exactly identifying $\pi^*$ or collect some information from the annotators. This section focuses on the former, and the latter is the subject of Sec. 6. Next, we introduce an approximate alignment objective, along with the required annotator information and algorithms to solve it.

## 5.1 First-Order Approximation

The approximation in Eq. (12) is equivalent to using a zeroth-order Taylor expansion of $\sigma(\cdot)$ around the average reward to calculate the likelihood function. To improve it, we extend DPO by incorporating an additional non-zero term from the expansion, which we call *first-(non-zero)-order corrected DPO*. The derivation is as follows. Expanding $\sigma\big(\Delta r^*(\boldsymbol{y}_1, \boldsymbol{y}_2; u)\big)$ around $\Delta \bar{r}^*(\boldsymbol{y}_1, \boldsymbol{y}_2) := \mathbb{E}_u\big[\Delta r^*(\boldsymbol{y}_1, \boldsymbol{y}_2; u)\big]$ up to the second order gives the approximation below for the likelihood:

$$\mathbb{E}_u\Big[\sigma\big(\Delta r^*(\boldsymbol{y}_1, \boldsymbol{y}_2; u)\big)\Big] \approx \sigma\big(\Delta \bar{r}^*(\boldsymbol{y}_1, \boldsymbol{y}_2)\big) + \frac{1}{2}\sigma''\big(\Delta \bar{r}^*(\boldsymbol{y}_1, \boldsymbol{y}_2)\big) \cdot \mathrm{Var}_u\big[\Delta r^*(\boldsymbol{y}_1, \boldsymbol{y}_2; u)\big]. \quad (14)$$

Here, the approximation omits $O(\mathbb{E}_u[(\Delta r^* - \Delta \bar{r}^*)^3])$. We can see from this equation that the approximation in Eq. (12) is loose when $\sigma$ is nonlinear and preferences have high variance.

To calculate Eq. (14), we can substitute $\Delta \bar{r}^*$ in by the difference in log policy ratios (Eq. (11)). We then need to estimate the variance term. Sec. 5.3 offers a variance estimator. Once the variance is estimated by a function $V(\boldsymbol{y}_1, \boldsymbol{y}_2)$, first-order corrected DPO estimates $\mathrm{Pr}(\boldsymbol{y}_2 \succ \boldsymbol{y}_1 \mid \pi)$ using

$$\sigma\big(h(\boldsymbol{y}_1, \boldsymbol{y}_2; \pi)\big) + \frac{\alpha}{2}\,\sigma''\big(h(\boldsymbol{y}_1, \boldsymbol{y}_2; \pi)\big) \cdot V(\boldsymbol{y}_1, \boldsymbol{y}_2)\,.$$

Here, $\alpha > 0$ determines the strength of correction, and

$$h(\boldsymbol{y}_1, \boldsymbol{y}_2; \pi) := \beta \log \frac{\pi(\boldsymbol{y}_2)}{\pi_{\mathrm{ref}}(\boldsymbol{y}_2)} - \beta \log \frac{\pi(\boldsymbol{y}_1)}{\pi_{\mathrm{ref}}(\boldsymbol{y}_1)} \quad (15)$$

denotes the difference of $\pi$'s induced rewards. Given $\Pr(\boldsymbol{y}_2 \succ \boldsymbol{y}_1 \mid \pi)$ and a preference dataset $\mathcal{D}$, we can maximize the log-likelihood similar to Eq. (6). For numerical stability, we use a stable logarithm $\widetilde{\log}(z) := \log(\max\{z, \epsilon\})$ in computations. Note that our theory suggests $\alpha = 1$ while DPO uses $\alpha = 0$. Our empirical findings in Sec. 7.1 show that larger values of $\alpha$ improve the effectiveness of the correction. We next discuss the estimation of $V(\boldsymbol{y}_1, \boldsymbol{y}_2)$.

## 5.2 Impossibility without Annotator Information

If we limit our algorithms to M-estimators, which encompass most practical learning methods, consistent estimation of the variance term is impossible with anonymous data:

**Proposition 5.2.** *There exists no consistent M-estimator that without annotator information can estimate* $V(\boldsymbol{x}, \boldsymbol{y}_1, \boldsymbol{y}_2) := \mathrm{Var}_u\big[\Delta r^*(\boldsymbol{x}, \boldsymbol{y}_1, \boldsymbol{y}_2; u)\big]$.

See proof on page 28. While Prop. 5.1 already implies that $\pi^*$ is unlearnable without annotator information, Prop. 5.2 goes further, showing that even a first-order improvement to DPO is practically impossible. Next, we show how minimal annotation information can overcome this impossibility.

## 5.3 Using Paired Preferences

We can get around Prop. 5.2 by collecting minimal information on annotators. Consider a dataset $\mathcal{D}$ of pairs of preferences in the form $\{(\boldsymbol{x}, \boldsymbol{y}_1, \boldsymbol{y}_2, o), (\boldsymbol{x}', \boldsymbol{y}_1', \boldsymbol{y}_2', o')\}$ where $o = \mathbb{1}\{\boldsymbol{y}_2 \succ \boldsymbol{y}_1\}$ and $o' = \mathbb{1}\{\boldsymbol{y}_2' \succ \boldsymbol{y}_1'\}$ are labeled by the *same person*. Using $\mathcal{D}$, we can train a *joint likelihood model* $J(\boldsymbol{x}, \boldsymbol{y}_1, \boldsymbol{y}_2, \boldsymbol{x}', \boldsymbol{y}_1', \boldsymbol{y}_2')$ by minimizing cross-entropy between $J$ and $(o \cdot o')$ as the label. The joint likelihood model consistently estimates

$$J(\boldsymbol{x}, \boldsymbol{y}_1, \boldsymbol{y}_2, \boldsymbol{x}', \boldsymbol{y}_1', \boldsymbol{y}_2') = \mathbb{E}_u\Big[\sigma\big(\Delta r^*(\boldsymbol{x}, \boldsymbol{y}_1, \boldsymbol{y}_2; u)\big) \cdot \sigma\big(\Delta r^*(\boldsymbol{x}', \boldsymbol{y}_1', \boldsymbol{y}_2'; u)\big)\Big].$$

We next show that this is in fact sufficient to estimate the variance term:

**Lemma 5.3.** *Using $J_1$ and $J_2$ as shorthands for $J(\boldsymbol{x}, \boldsymbol{y}_1, \boldsymbol{y}_2, \boldsymbol{x}, \boldsymbol{y}_1, \boldsymbol{y}_2)$ and $J(\boldsymbol{x}, \boldsymbol{y}_1, \boldsymbol{y}_2, \boldsymbol{x}, \boldsymbol{y}_2, \boldsymbol{y}_1)$, we can use the following to consistently estimate the variance term:*

$$V(\boldsymbol{y}_1, \boldsymbol{y}_2) = \frac{J_1 - (J_1 + J_2)^2}{\sigma'\big(\Delta \bar{r}^*(\boldsymbol{y}_1, \boldsymbol{y}_2)\big)^2}. \tag{16}$$

See proof on page 28. Note that we can substitute $\Delta \bar{r}^*(\boldsymbol{y}_1, \boldsymbol{y}_2)$ in terms of log policy ratios from Eq. (11). Thus, we have all the elements to calculate $V$ in Eq. (16). This completes our derivation of first-order corrected DPO.

# 6 Direct Alignment with Maximum Annotator Information

Recall that learning the optimal policy $\pi^*$ from anonymous data is impossible, and an approximate improvement to DPO requires minimal information about the annotations. But what if we collect richer data? Can we design a direct alignment method that consistently learns the optimal policy? To explore this, we consider a dataset where every sample is labeled by representatives of all user types. We show that consistent direct alignment is possible in this setting but at the cost of sample efficiency.

Suppose user types are in a finite set $\mathcal{U}$ and equally represented. This assumption makes our negative results stronger. Consider a rich data collection: For every query and candidate responses $(\boldsymbol{x}, \boldsymbol{y}_1, \boldsymbol{y}_2)$, we collect one preference data point from each user type. Let $\boldsymbol{o} \in \{0, 1\}^{\mathcal{U}}$ be the vector that indicates preferences where $o_u = 1$ if $\boldsymbol{y}_2 \succ \boldsymbol{y}_1$ by a user of type $u$, and 0 otherwise. Given such a dataset $\mathcal{D}$ with query, responses, and preferences as $(\boldsymbol{x}, \boldsymbol{y}_1, \boldsymbol{y}_2, \boldsymbol{o})$, our goal is to design a loss

$$\mathcal{L}(\mathcal{D}; \pi) = \frac{1}{|\mathcal{D}|} \sum_{(\boldsymbol{x}, \boldsymbol{y}_1, \boldsymbol{y}_2, \boldsymbol{o}) \in \mathcal{D}} l(\boldsymbol{x}, \boldsymbol{y}_1, \boldsymbol{y}_2, \boldsymbol{o}; \pi), \tag{17}$$

such that $\arg\min_\pi \mathcal{L}(\mathcal{D}; \pi)$ is a consistent estimator of $\pi^*$. Designing such a loss is, in fact, possible. For instance, suppose we only look into the agreement cases in $\mathcal{D}$ where $\boldsymbol{o}$ is either all one or zero. Conditioned on agreement, we will show that the probability of $\boldsymbol{y}_2 \succ \boldsymbol{y}_1$ is proportional to $\exp\big(\sum_u \Delta r^*(\boldsymbol{y}_1, \boldsymbol{y}_2; u)\big)$. We can express this likelihood directly in terms of log policy ratios of $\pi^*$ (see Eq. (11)). We formally show this possibility through a temperature-adjusted DPO:

**Proposition 6.1.** *Defining $l$ in Eq.* (17) *as follows results in a consistent estimation of the optimal policy when preferences follow the BT model:*

$$l(\boldsymbol{y}_1, \boldsymbol{y}_2, \boldsymbol{o}; \pi) = \begin{cases} -\log \sigma\big(|\mathcal{U}| \cdot h(\boldsymbol{y}_1, \boldsymbol{y}_2; \pi)\big), & \boldsymbol{o} = \mathbf{1}, \\ -\log \sigma\big(|\mathcal{U}| \cdot h(\boldsymbol{y}_2, \boldsymbol{y}_1; \pi)\big), & \boldsymbol{o} = \mathbf{0}, \\ 0 & \textit{o.w.} \end{cases}$$

*Here, $h$ is the difference of $\pi$'s induced rewards (Eq.* (15)*), and $\mathbf{1}$ ($\mathbf{0}$) is the vector of all ones (zeros).*

See proof on page 29. Consistent loss functions are not unique. We give another example in Prop. F.3, and a systematic way to find such losses in Lemma G.2. In both examples, loss functions reduce to the standard DPO loss when $|\mathcal{U}| = 1$.

While the loss function in Prop. 6.1 benefits from consistency, it only uses samples where all user types have agreed. In other words, it discards a sample with any disagreement. A natural question arises: Can we design a loss function that uses all data, including those with disagreement, while maintaining consistency? Surprisingly, the answer is no:

**Theorem 6.2.** *Suppose $l$ in Eq.* (17) *only depends on $(\boldsymbol{x}, \boldsymbol{y}_1, \boldsymbol{y}_2)$ through $\pi$ and $\pi_{\mathrm{ref}}$. If there are more than three types of user and the preferences follow BT, any loss that allows a consistent estimation of the optimal policy discards samples with disagreement, i.e., those with $\boldsymbol{o} \notin \{\mathbf{0}, \mathbf{1}\}$.*

See proof on page 29. This theorem highlights a tension: To improve efficiency, one must compromise either consistency or direct optimization. The approximate direct alignment method proposed in Sec. 5 exemplifies forgoing consistency. Next, we discuss an alternative that favors consistency.

**An Indirect Practical Solution: Averaging Personalized Rewards.** The tradeoff between sample efficiency and consistency arises from the requirement for direct optimization. To regain sample efficiency, we may relax the requirement for direct alignment by training reward models while still avoiding RL. Specifically, we can learn personalized reward models $r(\cdot; u)$ for different user types $u \in \mathcal{U}$, calculate a user-weighted expected reward, and use it to relabel a preference dataset. A dataset labeled with this average reward makes any direct alignment method applicable and is found effective in practice [45]. It is both consistent and sample-efficient when personalized reward learning is feasible, but comes at the cost of additional training and memory for each user type.

# 7 Experiments

We provide empirical evidence for our claims throughout the paper. We first extend our sensitivity example in Sec. 4.3 to a real-world preference dataset in Fig. 4. Using 19 Pew surveys, we find that NBC rankings change under shifts from uniform sampling in $20\%$ of cases. Notably, these changes require only modest shifts: In half the cases, a total variation distance of less than $0.23$ from uniform is enough to alter the rankings. We defer the details to Appendix B.

In Sec. 7.1, we simulate DPO and our proposed improvements in a small-scale environment where we can visualize and compare the resulting policies. Finally, we scale this experiment in Sec. 7.2 by fine-tuning large language models, illustrating the extent of possible improvement over DPO.

Code for reproducing all results is publicly available at `https://github.com/arashtne/dahp`.

## 7.1 Synthetic Experiments

We generalize the discrete environment of Xu et al. [46] with multiple user types. This enables us to visualize the differences between DPO's policy and the optimal policy, as well as to evaluate the effectiveness of applying a first-order correction (Sec. 5) and using a consistent loss function (Sec. 6).

**Environment.** A prompt $x$ can take a value from $1$ to $n$. There are also only $n$ possible responses to each $x$. The reward for responding $y$ to $x$ for a type $u$ is $r^*([x, y]; u) = R_u(\mathrm{dist}(x + u, y))$, where $\mathrm{dist}$ is a circular distance, and $R_u$ is a linearly decreasing function floored at zero. We set $n = 40$ and consider three equally represented types $\mathcal{U} = \{-10, 0, 10\}$ with BT annotators.

Since the reward (and thus the policies) depends only on $y - x$, we can reduce everything to a 1D representation by setting $\delta := y - x$ and averaging over $x$. For example, for a policy $\pi$, define a 1D

policy $\pi(\delta) := \frac{1}{n} \sum_{x \in [n]} \pi(x + \delta \mid x)$. We also compute standard errors of these 1D representations across $x$. Fig. 2(a) shows our choice of rewards as well as the expected reward across user types.

**Policies.** For a uniform $\pi_{\mathrm{ref}}$ and $\beta = 1$, Eq. (10) implies $\pi^*(y \mid x) \propto \exp\left(\frac{1}{3} \sum_{u \in \mathcal{U}} r^*([x, y]; u)\right)$. We generate a large dataset of preferences under uniform context and alternative distributions and use the Adam optimizer to minimize the loss for different methods. For the first-order correction of DPO, we additionally train a joint likelihood model $J$ to estimate the variance term $V$ from Eq. (16). We use the loss from Prop. 6.1 as our choice for the consistent loss.

**Results.** Fig. 2(b) presents $\pi_{\mathrm{DPO}}$ along with $\pi^*$ and NBC. Unlike the optimal policy which prefers $\delta$ around $-10$ and $10$, DPO prefers $\delta \approx 0$, and to a large extent is ordinally consistent with NBC.

Fig. 2(c) shows that increasing correction strength $\alpha$ brings the corrected DPO policy closer to $\pi^*$. In particular, at $\alpha = 1$, the corrected DPO already favors alternatives with $\delta \in \{-10, 10\}$, consistent with $\pi^*$. Furthermore, increasing $\alpha$ makes these alternatives even more favorable. In the full-information setting, Fig. 2(d) shows that minimizing the consistent loss largely leads to $\pi^*$. Note that minor deviations from theoretical derivations are likely due to limited data and imperfect optimization.

We have also simulated noisy annotations, where with a small probability, a random user provides the annotation instead of the intended user type (see Appendix C). Our results show that consistent loss minimization is largely robust to such noise, although this does not address its sample inefficiency in settings with many user types.

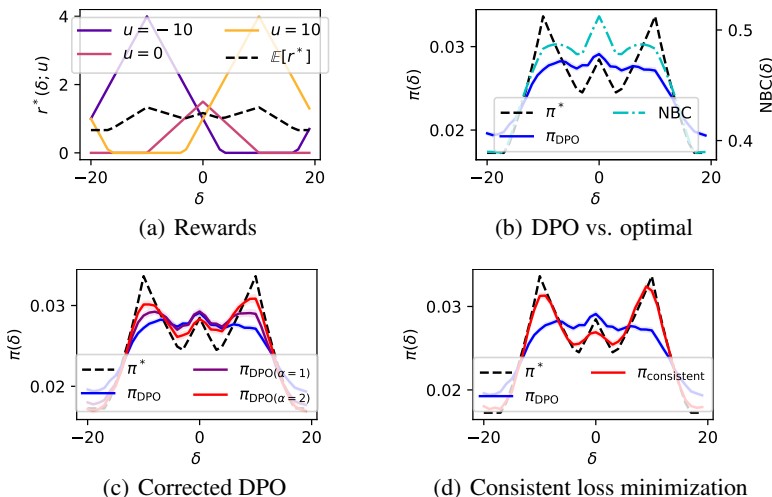

Figure 2: Rewards and policies in the synthetic setup.

### 7.2 Semi-Synthetic Experiments

While we have shown that DPO does not maximize the (user-weighted) average reward, the extent of its deviation from the optimal policy remains unclear. To assess the potential gains from accounting for heterogeneity, we compare DPO with a policy that minimizes the consistent loss function of Prop. 6.1 in a more realistic setup. Specifically, we use LoRA [47] to fine-tune Llama-3-8B [48] for both reward learning and direct alignment on two relabeled variations of the HH-RLHF dataset [49]. To simulate heterogeneous preferences, we define three user types with distinct length-based rewards (see Appendix D for details and similar results using Qwen-2.5-7B).

**Results.** We use agreement with the ground-truth average reward on the test set as the success metric. Using standard reward learning and DPO on an anonymous preference dataset—where each sample is labeled by a random user type—the induced ranking agrees with the average reward in $89.6\%$ and $67.4\%$ of test cases, respectively (Fig. 3, blue). When we use a consistent loss with full annotator information, agreement improves to $93.9\%$ for reward learning and $71.7\%$ for direct alignment (Fig. 3, red). In summary, explicitly accounting for heterogeneity improves agreement with the average reward by $4.3$ percentage points, highlighting the substantial room for improvement left by DPO.

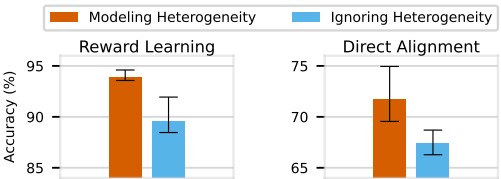

Figure 3: In the presence of preference labels from every user type, our proposed loss function produces reward models (left) and aligned policies (right) that are more consistent with the average reward across user types, compared to typical approaches that overlook heterogeneity. Bars show the mean, and whiskers denote the second and third quartiles across five random seeds.

# 8 Related Work

Aligning models to "serve pluralistic human values" [25] can involve personalization to the user's specific reward [12–17] or aggregation of diverse rewards [18, 19]. The latter, which is the subject of our study, can use insights from social choice theory [24, 23].

Closest to our work, EM-DPO [18] simultaneously learns the distribution of user types and their corresponding policies. However, EM introduces significant complexities and lacks formal guarantees. Moreover, the identifiability of types requires additional assumptions. MODPO [50] applies DPO for each user type while utilizing estimated rewards from other user types to maximize a linear combination of rewards. Neither method obtains a policy by directly minimizing a loss over preference data. For a more extensive review of related work, refer to Appendix E.

# 9 Discussion and Conclusion

Aligning a single policy to the average reward across user types requires collecting annotator information. This can range from minimal information such as linking two instances labeled by the same annotator, to richer information like using questionnaires to infer annotator types. We improved DPO using the former and introduced a consistent loss when annotators from all user types label every data point. With additional assumptions, unsupervised methods might be able to identify annotator types from anonymous datasets [51]. Further research should explore the additional structures that, when used during data collection, can help with identifiability.

Our results revealed a tension between consistency and sample efficiency in direct alignment. Thus, an alternative approach, i.e., individual reward training and aggregation, may be more practical for addressing heterogeneity when individual rewards are identifiable.

We use the average reward as the natural choice among aggregations that give a well-defined alignment problem. However, the choice of aggregation is inherently a social and policy question rather than purely a technical one. In certain contexts, the policymaker might prefer to give higher weight to disadvantaged people to address issues such as inequality. Additionally, in some cases, the very existence of a reward function may be questionable, requiring objectives to be defined in terms of choice probabilities rather than rewards.

We believe that trained policies should not be used to elicit or represent aggregate preferences, even when reward aggregation is appropriate and estimation is consistent. While such policies may capture certain patterns in user behavior, they do not necessarily reflect the underlying interests or values of the population. In other words, we view the resulting policy as a functional tool for decision-making rather than a true representation of users' collective interests.

In summary, while preference heterogeneity is well recognized in mathematical psychology, standard methods often bypass this complexity. As we showed, accounting for heterogeneity, even when the goal remains the same as in the homogeneous setting—to derive a single policy— can render common techniques inefficient or inapplicable. Understanding these limitations calls for new approaches that explicitly incorporate heterogeneity while balancing efficiency, consistency, and practicality.

## Acknowledgment

We thank Mohammad Alizadeh, Pouya Hamadanian, Moritz Hardt, Erfan Jahanparast, and Itai Shapira for discussions and feedback on earlier versions of this paper. Abebe was partially supported by the Andrew Carnegie Fellowship Program. Procaccia was partially supported by the National Science Foundation under grants IIS-2147187 and IIS-2229881; by the Office of Naval Research under grants N00014-24-1-2704 and N00014-25-1-2153; and by a grant from the Cooperative AI Foundation.

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

# A Additional Drawbacks of DPO under Heterogeneity

**Violating Independence of Irrelevant Alternatives (IIA).** Suppose $\mathcal{U} = \{A, B\}$ and types are equally represented. Given two possible responses $\boldsymbol{y}_1$ and $\boldsymbol{y}_2$, type $A$ prefers $\boldsymbol{y}_1$ but type $B$ prefers $\boldsymbol{y}_2$:

$$r^*(\boldsymbol{y}_1; A) = 6, \, r^*(\boldsymbol{y}_2; A) = 1,$$
$$r^*(\boldsymbol{y}_1; B) = 3, \, r^*(\boldsymbol{y}_2; B) = 9.$$

A direct calculation shows $\mathbb{E}_u[r^*(\boldsymbol{y}_2)] > \mathbb{E}_u[r^*(\boldsymbol{y}_1)]$ and $\mathrm{NBC}(\boldsymbol{y}_2) > \mathrm{NBC}(\boldsymbol{y}_1)$. So, both $\pi^*$ and $\pi_{\mathrm{DPO}}$ prefer $\boldsymbol{y}_2$. Let's consider another possible response $\boldsymbol{y}_3$ which is not the preferred response for any user type:

$$r^*(\boldsymbol{y}_3; A) = r^*(\boldsymbol{y}_3; B) = 2.$$

While $\pi^*$ still prefers $\boldsymbol{y}_2$ to $\boldsymbol{y}_1$, now $\mathrm{NBC}(\boldsymbol{y}_1) \approx 0.62 > \mathrm{NBC}(\boldsymbol{y}_2) \approx 0.55$, so, introducing an irrelevant alternative can alter DPO's ranking over existing alternatives.

**Tyranny of Majority.** Suppose $\mathcal{U} = \{A, B\}$ with type $A$ shaping $90\%$ of the population. Given two responses $\boldsymbol{y}_1, \boldsymbol{y}_2$, type $A$ slightly favors $\boldsymbol{y}_2$ but type $B$ finds $\boldsymbol{y}_2$ offensive:

$$r^*(\boldsymbol{y}_1; A) = 0.5, \, r^*(\boldsymbol{y}_2; A) = 1,$$
$$r^*(\boldsymbol{y}_1; B) = 0.5, \, r^*(\boldsymbol{y}_2; B) = -10.$$

In this case, $\pi^*$ prefers $\boldsymbol{y}_1$ even though type $B$ is a minority. In contrast, we have $\mathrm{NBC}(\boldsymbol{y}_1) \approx 0.47, \mathrm{NBC}(\boldsymbol{y}_2) \approx 0.53$, which implies that the majority dominates in DPO.

# B PEW Surveys Experiments: Details and Additional Examples

In this section, we expand on our PEW surveys experiment where we used polling data on key political and social issues to show: (i) how NBC rankings can differ from those maximizing the average reward; (ii) How sensitive NBC is to the sampling distribution of the pairwise preference data.

**Data.** We use several Pew Research Center surveys, specifically the American Trends Panel surveys number 35, 52, 79, 83, 99, 109, 111, 112, 114, 119, 120, 121, 126, 127, 128, 129, 130, 131, and 132. The choices are a mix of recent surveys and those relevant to science, technology, data and AI. Each survey include questions asked to thousands of participants. We categorize participants by political party leanings to define types. When processing the questions, we discard responses that are empty, as well as discarding the option "Refused". We note that discarding the option "Refused" had no effect on the results as it is not frequently chosen.

**Reward Estimation.** Although we observe how often each group selects a particular option, we don't directly observe respondents' internal rewards. To estimate this, we apply the Luce-Shepherd model [39, 32]:

$$\Pr\left(\text{option } i \text{ is chosen from } \mathcal{S}\right) = \frac{\exp\left(r(i; u)\right)}{\sum_{j \in \mathcal{S}} \exp\left(r(j; u)\right)}, \tag{18}$$

where $\mathcal{S}$ is the set of options, and $r(\cdot; u)$ is the reward for type $u$. This allows us to estimate each option's reward (up to a constant additive term) for each type. From these estimates and observed probabilities, we compute both the expected reward and the NBC metric, where in the latter we assume the uniform probability for alternatives unless specified otherwise.

**NBC Sensitivity to Sampling Distribution: Results.** Recall from Sec. 4.2 that the common practice of alignment assuming homogeneity results in ordinal consistency with NBC. Here, we analyze the sensitivity of NBC to the distribution of pairwise preference datasets in real-world cases by using Pew surveys [52], the same dataset used in Sec. 4.3. Specifically, we address two questions: (i) Across the questions in the Pew surveys, how often would NBC rankings change when the sampling distribution of alternatives varies (while retaining support over all alternatives)? (ii) How much must the sampling distribution deviate from uniform to alter NBC rankings?

To answer these questions, we first estimate the reward of each option in each question (see Sec. 4.3 and Appendix B for further details). Given the rewards, we can calculate NBC under any sampling distribution using Eq. (13).

For question (i), among 1519 questions from 19 Pew surveys, NBC rankings change due to changing the sampling distribution from uniform in 20% of cases (306 questions), with the preferred choice changing in 136 cases. For question (ii), we find that a modest change in the sampling distribution suffices; in half the cases, a total variation (TV) distance of less than 0.23 from uniform alters the rankings. The cumulative distribution function (CDF) of the minimum TV distances required to change NBC rankings is shown in Fig. 4. Further experimental details are in Appendix B.

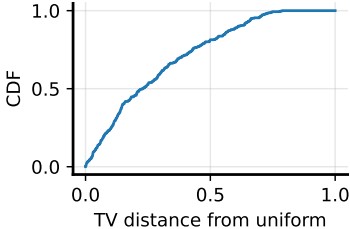

Figure 4: CDF of the minimum TV distances (from uniform) required to change the NBC order in the Pew surveys. A change of $0.23$ is sufficient to change the order in half the questions.

**NBC Sensitivity to Sampling Distribution: Implementation Details.** Above, discussed the sensitivity of NBC rankings to the sampling distribution of pairwise preference data. Here, we discuss the implementation details.

Recall that NBC is defined as:

$$\text{NBC}(\boldsymbol{y}; \mathcal{D}) := \mathbb{E}_{\boldsymbol{y}' \sim \mathcal{D}(\cdot)}\big[\Pr(\boldsymbol{y} \succ \boldsymbol{y}' \mid \boldsymbol{x}; r)\big].$$

To compute this, we first estimate the reward function $r$, then evaluate $\Pr(\boldsymbol{y} \succ \boldsymbol{y}'; r)$ for all $(\boldsymbol{y}, \boldsymbol{y}') \in \mathcal{Y} \times \mathcal{Y}$, where $\mathcal{Y}$ is the set of alternatives. Next, consider the feasibility of swapping the ranking induced by the uniform distribution $\mathcal{D}_U$ for alternatives $\boldsymbol{y}_i$ and $\boldsymbol{y}_j$ with a new distribution $\mathcal{D}_a$, assuming $\text{NBC}(\boldsymbol{y}_i; \mathcal{D}_U) > \text{NBC}(\boldsymbol{y}_j; \mathcal{D}_U)$. This is equivalent to solving the following linear program:

$$\begin{aligned}
\text{minimize} \quad & \frac{1}{2}\mathbf{1}^\top \mathbf{s}, \\
\text{subject to:} \quad & \mathbf{q} > \epsilon \mathbf{1}, \\
& \mathbf{s} \geq \frac{1}{N}\mathbf{1} - \mathbf{q}, \\
& \mathbf{s} \geq \mathbf{q} - \frac{1}{N}\mathbf{1}, \\
& \mathbf{P}_i \mathbf{q} < \mathbf{P}_j \mathbf{q} + \delta, \\
& \mathbf{1}^\top \mathbf{q} = 1, \\
& \mathbf{q} > \mathbf{0}.
\end{aligned}$$

Here, $N = |\mathcal{Y}|$, and labeling the alternatives as $\boldsymbol{y}_1, \ldots, \boldsymbol{y}_N$, we define $\mathbf{P}_{ij} = \Pr(\boldsymbol{y}_i \succ \boldsymbol{y}_j; r)$, $q_i = \mathcal{D}_a(\boldsymbol{y}_i)$, and $\mathbf{P}_k$ as the $k$-th row of $\mathbf{P}$. The parameter $\epsilon > 0$ ensures support for all alternatives, and $\delta > 0$ controls the required magnitude of change in NBC beyond what is required for the swap. We set $\epsilon = \delta = 10^{-5}$.

To compute the minimum TV distance, we solve the program for all pairs $(i, j)$ where $\text{NBC}(\boldsymbol{y}_i; \mathcal{D}_U) > \text{NBC}(\boldsymbol{y}_j; \mathcal{D}_U)$ and record the smallest objective value. In this analysis, we group respondents by political leaning (specifically, the column `F_PARTYSUM_FINAL`). We also note that in this analysis we exclude survey questions with fewer than three options.

**Additional Examples.** We highlight some of the examples of discrepancies that we find in some of the surveys listed above in Figures 5, 6, 7, 8, and 9. We note though that while we indeed find a few examples of the discrepancy, NBC rankings is actually aligned with average reward rankings in most cases. One possible explanation for this is that in many questions, the distributions of responses across types were very similar, suggesting that a homogeneous reward model would have been appropriate, causing NBC and average reward to align.

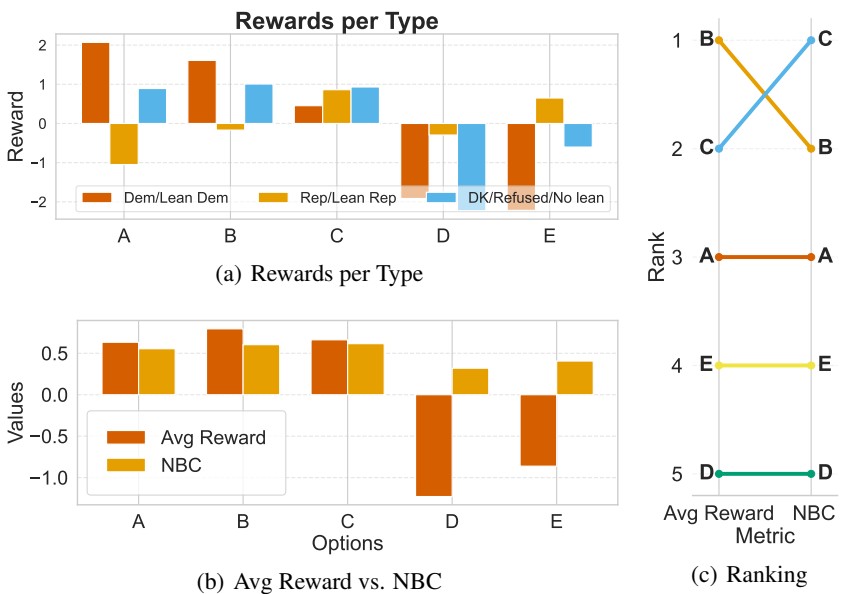

(a) Rewards per Type

(b) Avg Reward vs. NBC

(c) Ranking

Figure 5: Do you think the plans and policies of the Biden administration will make the country's response to the coronavirus outbreak: A: A lot better; B: A little better; C: Not much different; D: A little worse; E: A lot worse

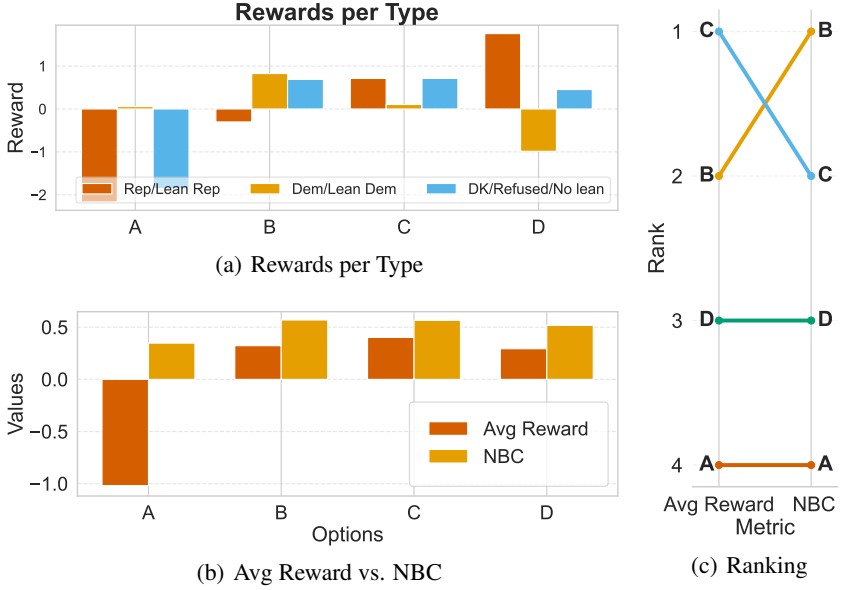

(a) Rewards per Type

(b) Avg Reward vs. NBC

(c) Ranking

Figure 6: How would you rate the job Joe Biden is doing responding to the coronavirus outbreak? A: Excellent; B: Good; C: Only fair; D: Poor

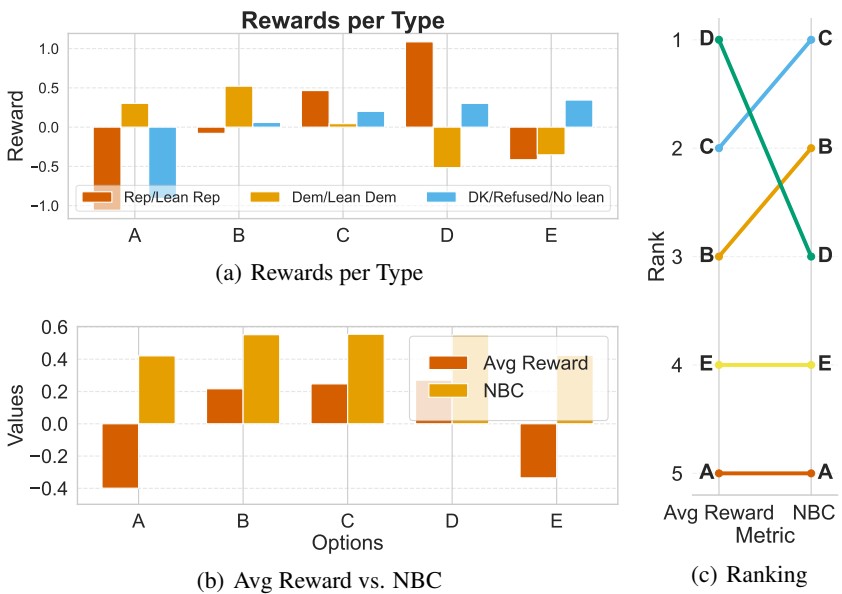

(a) Rewards per Type

(b) Avg Reward vs. NBC

(c) Ranking

Figure 7: The next time you purchase a vehicle, how likely are you to seriously consider purchasing an electric vehicle? A: Very likely; B: Somewhat likely; C: Not too likely; D: Not at all likely; E: I do not expect to purchase a vehicle

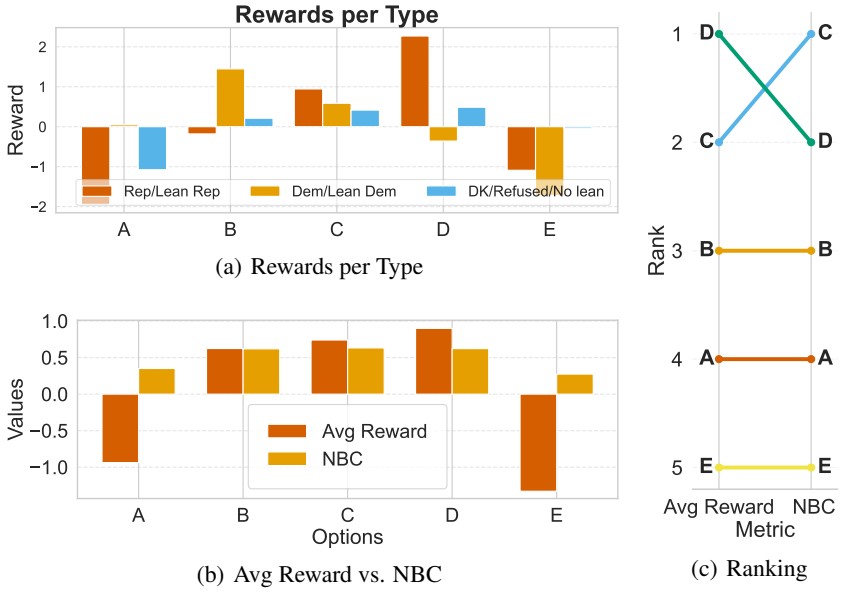

(a) Rewards per Type

(b) Avg Reward vs. NBC

(c) Ranking

Figure 8: What is your overall opinion of Kamala Harris? A: Very favorable; B: Mostly favorable; C: Mostly unfavorable; D: Very unfavorable; E: Never heard of this person.

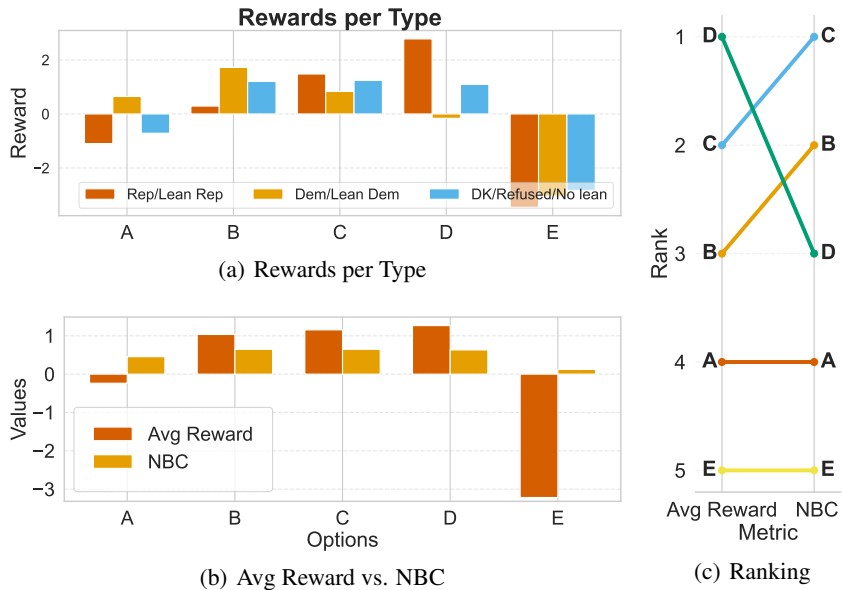

(a) Rewards per Type

(b) Avg Reward vs. NBC

(c) Ranking

Figure 9: What is your overall opinion of Joe Biden? A: Very favorable; B: Mostly favorable; C: Mostly unfavorable; D: Very unfavorable; E: Never heard of this person.

## C   Robustness to Noisy Preferences

We extend our analysis by simulating noisy user preferences in the synthetic setup of Sec. 7.1. For a noise level $n \leq 1$, each type-$u$ user's preference is replaced by that of a random user type with probability $n$. We compare the learned policies under varying noise levels with the optimal policy (OPT), reporting total variation (TV), Kendall's $\tau$, and Spearman's $r$.

As shown in Table 1, the TV between the noisy $\pi_{\text{consistent}}$ and OPT drops below that of DPO and OPT only at high noise levels ($n \geq 0.7$). Even in these cases, $\pi_{\text{consistent}}$ remains ordinally closer to OPT than DPO's policy, suggesting that the benefits of consistency may be robust to annotation noise.

We note, however, that this result is limited to synthetic settings and does not address the sample inefficiency of direct alignment methods which our key impossibility result still discourages their use in settings with many user types.

Table 1: Comparison of DPO and $\pi_{\text{consistent}}$ under different noise levels in the synthetic environment (Sec. 7.1).

| Metric | DPO | $\pi_{\text{consistent}}$ under noise level $n$ | | | | | | | |
|---|---|---|---|---|---|---|---|---|---|
| | | 0.0 | 0.1 | 0.2 | 0.3 | 0.4 | 0.5 | 0.6 | 0.7 |
| TV | 0.089 | 0.023 | 0.044 | 0.066 | 0.073 | 0.081 | 0.084 | 0.088 | 0.090 |
| Kendall's $\tau$ | 0.526 | 0.933 | 0.888 | 0.757 | 0.710 | 0.668 | 0.649 | 0.618 | 0.563 |
| Spearman's $r$ | 0.676 | 0.989 | 0.976 | 0.912 | 0.863 | 0.814 | 0.811 | 0.762 | 0.727 |

# D  Semi-Synthetic Experiments: Fine-Tuning Llama-3-8B and Qwen-2.5-7B on HH-RLHF

**Reward Models.** Fig. 10 shows the three distinct rewards we use for the three user types along with their average. In order to have a reliable ground-truth reward which we can rely on in evaluation, we define these rewards as functions of the number of tokens in prompt-response combinations.

**Anonymous Dataset.** We use prompts and response pairs from both helpfulness and harmlessness subsets of Anthropic's HH-RLHF dataset [49] and relabel the *chosen* and *rejected* responses manually. We filter for data points in which the sum of the number of tokens in the prompt and the number of tokens in the longer response do not exceed $512$. This leaves us with $160,800$ training and $17,104$ test data points. For every data point (a prompt with a pair of responses), we sample one of the three user types uniformly at random. Given the type of user, we sample a preference based on BT [34] to label the two alternatives.

**Dataset with Maximum Annotator Information.** We use prompts and response pairs from both helpfulness and harmlessness subsets of Anthropic's HH-RLHF dataset [49] and relabel the *chosen* and *rejected* responses manually. We filter for data points in which the sum of the number of tokens in the prompt and the number of tokens in the longer response do not exceed $512$. This leaves us with $160,800$ training and $17,104$ test data points. For every data point (a prompt with a pair of responses), we keep sampling BT [34] preferences from all user types until they agree with each other. Once the consensus is achieved, we stop sampling and use the agreed-upon preference as the label for this data point.

**Fine-Tuning Details.** We fine-tune Llama-3-8B [48] base model with LoRA [47]. We fine-tune for one epoch with a batch size of 2, and use a linear learning rate schedule that starts with $3 \times 10^{-5}$ and decreases to zero. We use the Adam optimizer with a weight decay of $0.001$ [53]. Regarding LoRA's hyper-parameters, we use the matrix rank of $r = 8$, $\alpha = 32$, and the dropout probability of $0.1$.

For direct alignment experiments, we use a uniform reference policy. When ignoring heterogeneity, we do vanilla DPO over the anonymous dataset. When modeling heterogeneity, we use the loss function we propose in Prop. 6.1 over the dataset with maximum annotator information. We use the ordinal agreement between the ground-truth average reward and the reward induced by the aligned policy as the measure of accuracy.

For the reward learning experiments, we fine-tune the Llama-3-8B as a reward model. When ignoring heterogeneity, we assume BT and maximize the probability of the anonymous preference dataset under the learned reward model. When modeling heterogeneity, we use the loss function in Prop. 6.1 over the dataset with maximum annotator information, but replace $h(\boldsymbol{y}_1, \boldsymbol{y}_2; \pi)$ with the difference in rewards, i.e., $r(\boldsymbol{y}_2) - r(\boldsymbol{y}_1)$. We use the ordinal agreement between the ground-truth average reward and the learned reward as the measure of accuracy.

**Detailed Results.** We conduct every experiment with five different random seeds. Fig. 3 shows the average, $25^{\text{th}}$ percentile, and $75^{\text{th}}$ percentile of accuracy across the five random seeds. Tables 2 and 3 show the raw accuracy numbers across the five random seeds.

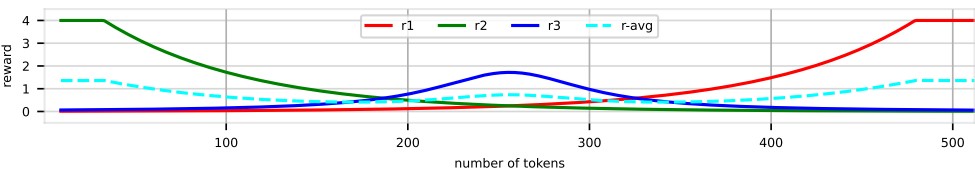

Figure 10: Reward definition for three user types in semi-synthetic experiments (Sec. 7.2) based on the length of prompt response combination. The first user type prefers long prompt response combinations, the second user type prefers short prompt response combinations, and the third user type prefers mid-length prompt response combinations. The dashed cyan line shows the average reward across the three user types.

Table 2: Accuracy (%) in Alignment Experiments (Llama-3-8B)

| SEED | IGNORING HOMOGENEITY | MODELING HETEROGENEITY |
|------|----------------------|------------------------|
| 0    | 65.61                | 66.63                  |
| 1    | 67.55                | 69.55                  |
| 2    | 68.77                | 75.21                  |
| 3    | 68.70                | 72.04                  |
| 4    | 66.28                | 74.95                  |

Table 3: Accuracy (%) in Reward Learning Experiments (Llama-3-8B)

| SEED | IGNORING HOMOGENEITY | MODELING HETEROGENEITY |
|------|----------------------|------------------------|
| 0    | 92.33                | 95.26                  |
| 1    | 85.38                | 92.01                  |
| 2    | 88.46                | 94.6                   |
| 3    | 89.69                | 93.57                  |
| 4    | 91.94                | 93.87                  |

Table 4: Accuracy (%) in Alignment Experiments (Qwen-2.5-7B)

| SEED    | IGNORING HOMOGENEITY | MODELING HETEROGENEITY |
|---------|----------------------|------------------------|
| 0       | 68.66                | 72.1                   |
| 1       | 63.68                | 69.77                  |
| 2       | 62.48                | 68.87                  |
| 3       | 63.80                | 71.57                  |
| 4       | 60.06                | 66.72                  |
| AVERAGE | 63.74                | **69.81**              |

**Fine-Tuning Qwen-2.5-7B.** Using the same setup, we repeat the alignment experiment with Qwen-2.5-7B. As shown in Table 4, incorporating annotator information and explicitly modeling heterogeneity in user types—via the loss function introduced in Sec. 6—improves average accuracy by 6%.

# E   Additional Related Work

The challenge of handling heterogeneous preferences in alignment has been recognized as a significant problem in alignment research [54, 55, 24, 25]. This problem has attracted considerable attention from researchers in the field. Here, we highlight a few representative works that address key directions in tackling this challenge.

**Analysis of DPO.** Our study of how standard preference learning methods, such as DPO, behave in the presence of heterogeneous preferences was inspired by Siththaranjan et al. [20]'s result, which shows that RLHF aggregates preferences according to a well-known voting rule called Borda count. Chakraborty et al. [19] highlights the impossibility of aligning with a singular reward model in RLHF by providing a lower bound on the gap between the optimal policy and a subpopulation's optimal policy. Dumoulin et al. [26] adopts a density estimation perspective on learning from human feedback to illustrate the challenges of preference learning from a population of annotators with diverse viewpoints. Rosset et al. [56] and Gao et al. [57] point out the limitations of point-wise reward models in expressing complex, intransitive preferences that may arise due to the aggregation of diverse preferences. Additionally, frameworks that generalize DPO and unify different alignment methods have been proposed to analyze current approaches and explore possible alternatives [58, 59, 42, 60].

**Policy Personalization.** Many works in the literature have proposed personalization as a solution to the problem of pluralistic alignment. Poddar et al. [61] propose a latent variable formulation of the

problem and learn rewards and policies conditioned on it. Chen et al. [62] use an ideal point model for preferences and learn latent spaces representing different preferences. Mapping user information to user representations, Li et al. [13] perform personalized DPO to jointly learn a user model and a personalized language model. Balepur et al. [17] use abductive reasoning to infer user personas and train models to tailor responses accordingly. Lee et al. [14] explore the possibility of steering a language model to align with a user's intentions through system messages. Dang et al. [63] extend personalized alignment to text-to-image diffusion models. Jang et al. [12] perform personalized alignment by decomposing preferences into multiple dimensions. Lau et al. [15] dynamically adapt the model to individual preferences using in-context learning.

**Preference Aggregation.** Closely aligned with our goal of serving the entire population with a single policy, several works have explored ways to aggregate diverse preferences. The rich literature on social choice theory has proven to be a valuable source of inspiration for studying existing preference learning approaches and proposing new ones [23, 64, 65, 24, 66]. Drawing insights from social choice theory, robustness to approximate clones has been proposed as a desirable property of RLHF algorithms, which current methods lack [41]. The Minimax Winner, a concept in preference aggregation, has inspired the use of the proportion of wins as the reward for a particular trajectory to align a model through self-play [67]. The impact of heterogeneity on strategic behavior in feedback and its effects on aggregation are also explored in Park et al. [11], which further examines the use of different social welfare functions for preference aggregation.

**Methods.** Solutions proposed to address different formulations of the problem span a wide range of methods. Siththaranjan et al. [20] estimate a distribution of scores for alternatives to account for heterogeneity as hidden context. Chidambaram et al. [18] propose an Expectation-Maximization (EM) version of DPO to minimize a notion of worst-case regret. Multi-objective reinforcement learning [68, 12] and its direct optimization variant [50] have also been proposed to align with diverse preferences. Wang et al. [69] train a multi-objective reward model to capture diverse preferences. Zhong et al. [70] use meta-learning to learn diverse preferences and aggregate them using different social welfare functions. Li et al. [71] design an optimal-transport-based loss to calibrate their model with the categorical distribution of preferences. Producing a Pareto front of models has also been explored as a solution. Boldi et al. [72] employ an iterative process to select solutions, while Rame et al. [73] interpolate the weights of independent networks linearly to achieve a Pareto-optimal generalization across preferences.

**Empirical Observations.** Empirical studies of alignment methods have had a significant impact on the study of preference learning. Zhang et al. [5] demonstrate the Bradley-Terry model's failure to distinguish between unanimous agreement among annotators and the majority opinion in cases of diverging user preferences. Chen et al. [74] show that RLHF and DPO struggle to improve ranking accuracy. Zeng et al. [75] study the role of model size and data size in the impact of diversified human preferences. Bansal et al. [76] demonstrate the significant influence of feedback protocol choice on alignment evaluation. Santurkar et al. [77] explore the opinions reflected by a language model, while Bakker et al. [78] investigate a language model's ability to generate consensus statements by training it to predict individual preferences. Jiang et al. [79] propose individualistic alignment to predict an individual's values, and Zollo et al. [80] introduce the PersonalLLM benchmark to measure a model's adaptation to a particular user's preferences.

# F    Additional Statements

**Proposition F.1.** *There exists a mixture of BTs that a single BT cannot represent.*

**Definition F.2** (Learnability). Denote by $\mathcal{D}_{r,\sigma}$ an i.i.d. sampled pairwise preference dataset labeled by random users with reward $r$ and preference model $\sigma$. Let $\bar{r}(\boldsymbol{y}) := \mathbb{E}_u[r(\boldsymbol{y};u)]$. We say that the ranking based on $\bar{r}$ is (weakly) learnable if, for some $\epsilon > 0$, there exists an algorithm with a bounded sample complexity $m$, such that for every reward $r$, when given a dataset $\mathcal{D}_{r,\sigma}$ of size $|\mathcal{D}_{r,\sigma}| \geq m(\epsilon, \bar{r})$, it outputs a ranking consistent with $\bar{r}$ with a probability at least $\epsilon$ above the chance level.

**Proposition F.3.** *Defining $l$ in Eq. (17) as follows results in a consistent estimation of the optimal policy when preferences follow the BT model:*

$$l(\boldsymbol{y}_1, \boldsymbol{y}_2, \boldsymbol{o}; \pi) = \begin{cases} -\sigma\big(h(\boldsymbol{y}_1, \boldsymbol{y}_2; \pi)\big) - I\big(\sigma\big(h(\boldsymbol{y}_1, \boldsymbol{y}_2; \pi)\big)\big), & \boldsymbol{o} = \mathbf{1}, \\ -\sigma\big(h(\boldsymbol{y}_2, \boldsymbol{y}_1; \pi)\big) - I\big(\sigma\big(h(\boldsymbol{y}_2, \boldsymbol{y}_1; \pi)\big)\big), & \boldsymbol{o} = \mathbf{0}, \\ 0 & \text{o.w.} \end{cases}$$

*Here, we define $I(\theta) := \int_1^\theta \left(\frac{1}{\theta'} - 1\right)^{|\mathcal{U}|} \mathrm{d}\theta'$, and $h$ is the difference of $\pi$'s induced rewards (Eq. (15)).*

See proof on page 31.

# G    Missing Proofs

**Proposition 3.1.** *Consider a differentiable aggregation function $f : \mathbb{R}^{\mathcal{U}} \to \mathbb{R}$. Suppose that for every reward $r$ consistent with the preference distribution—meaning $\Pr(\boldsymbol{y}_2 \succ \boldsymbol{y}_1 \mid r) = \Pr(\boldsymbol{y}_2 \succ \boldsymbol{y}_1 \mid r^*)$ for all prompts and responses $\boldsymbol{y}_1$, $\boldsymbol{y}_2$ in the response space $\mathcal{Y}$—the function $f\big((r(\boldsymbol{y}; u))_{u \in \mathcal{U}}\big)$ induces the same ordering over $\mathcal{Y}$. If $\{(r(\boldsymbol{y}; u))_{u \in \mathcal{U}} \mid \boldsymbol{y} \in \mathcal{Y}\}$ has non-empty interior, then $f$ must be affine.*

*Proof of Proposition 3.1.* First of all, if a reward function $r^*(\boldsymbol{y}; u)$ can explain the preferences of a user type $u$, any other reward function $r(\boldsymbol{y}; u) := r^*(\boldsymbol{y}; u) + c(u)$ induces the same preference distribution:

$$\Pr(\boldsymbol{y}_2 \succ \boldsymbol{y}_1 \mid r; u) = \sigma\big(r(\boldsymbol{y}_2; u) - r(\boldsymbol{y}_1; u)\big) = \sigma\big(r^*(\boldsymbol{y}_2; u) - r^*(\boldsymbol{y}_1; u)\big) = \Pr(\boldsymbol{y}_2 \succ \boldsymbol{y}_1 \mid r^*; u).$$

Therefore, $r^*$ is identifiable up to a bias term that can depend on the context and user type.

Denote all the true rewards from different user types by a vector $\boldsymbol{r}^*(\boldsymbol{y}) \in \mathbb{R}^{\mathcal{U}}$. Let $\mathcal{Y}$ be the set of possible responses and $\mathcal{R}^* = \{\boldsymbol{r}^*(\boldsymbol{y}) \mid \boldsymbol{y} \in \mathcal{Y}\} \subseteq \mathbb{R}^{\mathcal{U}}$ be the set of possible rewards, which by assumption has a non-empty interior. Consider a reward aggregation function $f : \mathbb{R}^{\mathcal{U}} \to \mathbb{R}$. Our observation above implies that $f(\boldsymbol{r}^*(\boldsymbol{y}) + \boldsymbol{c})$ should induce a consistent ranking over $\boldsymbol{y}$ for every $\boldsymbol{c} \in \mathbb{R}^{\mathcal{U}}$. This means that $\mathrm{sign}\{f(\boldsymbol{r}_2 + \boldsymbol{c}) - f(\boldsymbol{r}_1 + \boldsymbol{c})\}$ does not depend on $\boldsymbol{c}$. Hence, there exists a function $\psi$ such that

$$\psi(\boldsymbol{r}_2 - \boldsymbol{r}_1) = \mathrm{sign}\{f(\boldsymbol{r}_2 + \boldsymbol{c}) - f(\boldsymbol{r}_1 + \boldsymbol{c})\},$$

for every $\boldsymbol{c} \in \mathbb{R}^{\mathcal{U}}$ and $\boldsymbol{r}_1, \boldsymbol{r}_2 \in (\mathcal{R}^*)^2$.

Consider an arbitrary $\boldsymbol{r}_1 \in \mathcal{R}^*$. We claim $\nabla f(\boldsymbol{r}_1) = \nabla f(\boldsymbol{r}_2)$ for every $\boldsymbol{r}_2 \in \mathcal{R}^*$. The proof is by contradiction: Suppose $\nabla f(\boldsymbol{r}_1) \neq \nabla f(\boldsymbol{r}_2)$. Since $|\mathcal{U}| > 1$, there should exist $\boldsymbol{\Delta} \in \mathbb{R}^{\mathcal{U}}$ such that $\langle \boldsymbol{\Delta}, \nabla f(\boldsymbol{r}_1) \rangle \geq 0 > \langle \boldsymbol{\Delta}, \nabla f(\boldsymbol{r}_2) \rangle$ (a similar argument holds for $\langle \boldsymbol{\Delta}, \nabla f(\boldsymbol{r}_1) \rangle > 0 \geq \langle \boldsymbol{\Delta}, \nabla f(\boldsymbol{r}_2) \rangle$ which we skip for brevity). Define $\boldsymbol{r}_1' = \boldsymbol{r}_1 + \epsilon\boldsymbol{\Delta}$ and $\boldsymbol{r}_2' = \boldsymbol{r}_2 + \epsilon\boldsymbol{\Delta}$, for a sufficiently small $\epsilon > 0$. We know $\boldsymbol{r}_1'$ and $\boldsymbol{r}_2'$ are in $\mathcal{R}^*$ as $\mathcal{R}^*$ has a non-empty interior. In the limit of $\epsilon \to 0^+$, we have

$$\psi(\epsilon\boldsymbol{\Delta}) = \psi(\boldsymbol{r}_1' - \boldsymbol{r}_1) = \mathrm{sign}\{\epsilon\langle \boldsymbol{\Delta}, \nabla f(\boldsymbol{r}_1) \rangle\} = 1,$$
$$\psi(\epsilon\boldsymbol{\Delta}) = \psi(\boldsymbol{r}_2' - \boldsymbol{r}_2) = \mathrm{sign}\{\epsilon\langle \boldsymbol{\Delta}, \nabla f(\boldsymbol{r}_2) \rangle\} = -1,$$

which is a contradiction. Therefore, $\nabla f$ should be constant everywhere in $\mathcal{R}^*$, which means $f$ can only be an affine function in this domain. $\qquad\square$

**Proposition 4.2.** *Suppose responses to $\boldsymbol{x}$ in the preference dataset are drawn from $\mathcal{D}(\cdot|\boldsymbol{x})$ and labeled according to BT. In the population limit, where empirical averages converge to expectations, DPO's induced reward, or equivalently $\frac{\pi_{\mathrm{DPO}}(\cdot|\boldsymbol{x})}{\pi_{\mathrm{ref}}(\cdot|\boldsymbol{x})}$, has the same ordering over responses as $\mathrm{NBC}(\cdot \mid \boldsymbol{x})$.*[4]

*Proof of Proposition 4.2.* We start from DPO's objective in Eq. (6). For notational simplicity, we assume $\pi(\boldsymbol{y} \mid \boldsymbol{x})$ already contains a normalization by $\pi_{\mathrm{ref}}(\boldsymbol{y} \mid \boldsymbol{x})$. In the limit of many data points,

---

[4]We can view $\mathrm{NBC}(\boldsymbol{y} \mid \boldsymbol{x})$ as an aggregation of rewards at $\boldsymbol{y}$. One can verify that NBC meets the order consistency condition of Prop. 3.1. However, it uses the reward value at $\boldsymbol{y}' \neq \boldsymbol{y}$ to define the aggregated reward at $\boldsymbol{y}$ and thus does not fall under Prop. 3.1. In fact, this interdependency causes the issues we discuss Sec. 4.3.

we can rewrite DPO's objective as the minimization of a cross-entropy loss

$$\mathcal{L}_{\text{DPO}}(\pi) := -\mathbb{E}_{\boldsymbol{x},\boldsymbol{y},\boldsymbol{y}'}\left[\bar{\sigma}\big(\Delta r^*(\boldsymbol{x},\boldsymbol{y}',\boldsymbol{y})\big)\cdot\log\sigma\Big(\beta\log\frac{\pi(\boldsymbol{y}\mid\boldsymbol{x})}{\pi(\boldsymbol{y}'\mid\boldsymbol{x})}\Big)\right.$$
$$\left.+\Big(1-\bar{\sigma}\big(\Delta r^*(\boldsymbol{x},\boldsymbol{y}',\boldsymbol{y})\big)\Big)\cdot\log\Big(1-\sigma\Big(\beta\log\frac{\pi(\boldsymbol{y}\mid\boldsymbol{x})}{\pi(\boldsymbol{y}'\mid\boldsymbol{x})}\Big)\Big)\right],$$

where $\bar{\sigma}\big(\Delta r^*(\boldsymbol{x},\boldsymbol{y}',\boldsymbol{y})\big)$ is shorthand for $\Pr(\boldsymbol{y}\succ\boldsymbol{y}'\mid\boldsymbol{x};r^*) = \mathbb{E}_u\big[\sigma\big(r^*([\boldsymbol{x},\boldsymbol{y}];u) - r^*([\boldsymbol{x},\boldsymbol{y}'];u)\big)\big]$. The minimizer of $\mathcal{L}_{\text{DPO}}$ should meet the first-order condition: $\frac{\partial\mathcal{L}_{\text{DPO}}}{\partial\pi(\boldsymbol{y}\mid\boldsymbol{x})} = 0$, for every $\boldsymbol{x}$ and $\boldsymbol{y}$. Then, a direct calculation shows that the optimal policy $\pi^*$ meets

$$\mathbb{E}_{\boldsymbol{y}'\sim\mathcal{D}(\cdot\mid\boldsymbol{x})}\left[\sigma\Big(\beta\log\frac{\pi^*(\boldsymbol{y}\mid\boldsymbol{x})}{\pi^*(\boldsymbol{y}'\mid\boldsymbol{x})}\Big)\right] - \mathbb{E}_{\boldsymbol{y}'\sim\mathcal{D}(\cdot\mid\boldsymbol{x})}\left[\bar{\sigma}\big(\Delta r^*(\boldsymbol{x},\boldsymbol{y}',\boldsymbol{y})\big)\right] = 0. \tag{19}$$

Recognize that the second term is $\text{NBC}(\boldsymbol{y}\mid\boldsymbol{x})$:

$$\text{NBC}(\boldsymbol{y}\mid\boldsymbol{x}) = \mathbb{E}_{\boldsymbol{y}'\sim\mathcal{D}(\cdot\mid\boldsymbol{x})}\left[\mathbb{E}_u\left[\sigma\big(r^*([\boldsymbol{x},\boldsymbol{y}];u) - r^*([\boldsymbol{x},\boldsymbol{y}'];u)\big)\right]\right].$$

In the absence of heterogeneity, we have $\bar{\sigma} = \sigma$, so setting $\beta\log\pi^*(\boldsymbol{y}\mid\boldsymbol{x}) = r^*([\boldsymbol{x},\boldsymbol{y}]) + C(\boldsymbol{x})$ for a normalizing $C$ would solve Eq. (19). In general, we are not aware of any closed-form solution. However, we can still infer the ordering the optimal policy induces from Eq. (19): Since the first term is increasing in $\pi^*(\boldsymbol{y}\mid\boldsymbol{x})$, the optimal policy will be monotone in $\text{NBC}(\boldsymbol{y}\mid\boldsymbol{x})$. This completes the proof. $\qquad\square$

**Proposition 5.1.** *When the preference model $\sigma(\cdot)$ is continuous, the ranking based on the (user-weighted) expected reward is not learnable (according to Def. F.2) without annotator information.*

*Proof of Proposition 5.1.* We give three related proof strategies. The first strategy works for every preference model. The second strategy draws on a connection to a robust version of Arrow's impossibility theorem [44]. The third strategy is inspired by Procaccia et al. [41]. We start with the notation and definitions specific to this proof.

**Notation and Definitions.** Consider a fixed prompt $\boldsymbol{x}$ with a set of possible responses $\mathcal{Y}$. Let $R$ denote a *complete ranking* over $\mathcal{Y}$, where $\boldsymbol{y}_2 R\boldsymbol{y}_1$ indicates whether $\boldsymbol{y}_2\succ\boldsymbol{y}_1$ or vice versa. A *profile* refers to a set of complete rankings. For a heterogeneous reward function $r(\boldsymbol{y};u)$ and a prior $\mathcal{P}$ over user types $\mathcal{U}$, let $R_{\bar{r}}$ be the ranking according to $\bar{r}(\boldsymbol{y}) := \mathbb{E}_{u\sim\mathcal{P}}[r(\boldsymbol{y};u)]$.

A *pairwise preference dataset* $\mathcal{D}$ consists of tuples $(\boldsymbol{y}_1,\boldsymbol{y}_2,o)$, where $o := \mathbb{1}\{\boldsymbol{y}_2\succ\boldsymbol{y}_1\}$. We assume that $\boldsymbol{y}_1$ and $\boldsymbol{y}_2$ are i.i.d. draws. When a random user with a reward function $r$ labels each instance in the dataset, we denote the resulting dataset by $\mathcal{D}_r$. A *pairwise learning algorithm* $\mathcal{A}$ produces a complete ranking over $\mathcal{Y}$ based on the pairwise preference dataset $\mathcal{D}$.

**Proof Strategy 1.** Suppose there exists an algorithm $\mathcal{A}$ such that for some $\mathcal{Y}$ with $|\mathcal{Y}|\geq 2$, for any reward function $r$ and any preference dataset $\mathcal{D}_r$ with $|\mathcal{D}_r|\geq n_{\bar{r}}$, it outputs $R_{\bar{r}}$ on $\mathcal{Y}$ with a probability of at least $\frac{1}{|\mathcal{Y}|!}+\epsilon$.

Suppose $r$ is a heterogeneous reward function that its expectation induces a complete ranking $R_{\bar{r}}$ with no tie. Define a new heterogeneous reward function $r_\gamma$ as follows. Consider a new user type $0\notin\mathcal{U}$. For some $\gamma > 1$, let $r_\gamma(\boldsymbol{y};u) = \gamma\,r(\boldsymbol{y};u)$ when $u\neq 0$, and $r_\gamma(\boldsymbol{y};0) = 0$. Define a new user distribution $\mathcal{P}_\gamma(u) := (1-\frac{1}{\gamma})\mathbb{1}\{u=0\} + \frac{1}{\gamma}\mathcal{P}(u)$. It is straightforward to verify $\bar{r}_\gamma := \mathbb{E}_{u\sim\mathcal{P}_\gamma}[r_\gamma(\boldsymbol{y};u)] = \bar{r}$. Therefore, with high probability, $\mathcal{A}$ outputs $R_{\bar{r}_\gamma} = R_{\bar{r}}$ from $\mathcal{D}_{r_\gamma}$ for every $\gamma > 1$:

$$\Pr\big(\mathcal{A}(\mathcal{D}_{r_\gamma}) = R_{\bar{r}}\big)\geq\frac{1}{|\mathcal{Y}|!}+\epsilon.$$

As we increase $\gamma$, for any continuous preference model $\sigma$, the pairwise preference dataset $\mathcal{D}_{r_\gamma}$ approaches a uniform preference dataset $\mathcal{D}_{\text{unif}}$ labeled mostly by an indifferent annotator of type $u = 0$. So, we have

$$\Pr\big(\mathcal{A}(\mathcal{D}_{\text{unif}}) = R_{\bar{r}}\big)\geq\frac{1}{|\mathcal{Y}|!}+\epsilon. \tag{20}$$

This is true for every $r$. For different choices of $r$, agreements with $R_{\bar{r}}$ are disjoint events. Since there are $|\mathcal{Y}|!$ different rankings overall, the pigeonholed principle implies $\epsilon = 0$.

**Proof Strategy 2.** The proof is by contradiction. Suppose there exists an algorithm $\mathcal{A}$ that for any reward function $r$ and any preference dataset $\mathcal{D}_r$ with $|\mathcal{D}_r| \geq n_{\bar{r}}$, it outputs $R_{\bar{r}}$ with a probability of at least $1 - \epsilon$. We follow Friedgut et al. [44] and define a social choice function as a function that yields an asymmetric relation on the alternatives given a profile. A social choice is *rational* if it is an order relation on the alternatives, and is *neutral* if it is invariant under permutations of alternatives.

Let $\mathcal{Y}_3 = \{\boldsymbol{y}_1, \boldsymbol{y}_2, \boldsymbol{y}_3\}$ be an arbitrary subset of $\mathcal{Y}$ with size 3. Next, we construct a neutral social choice function $f$ acting on $\mathcal{Y}_3$ that is independent of irrelevant alternatives (IIA). Here is how we design $f$: Let $P_r$ be a profile of size $n \geq n_{\bar{r}}$ at the input, where every $R \in P_r$ is an i.i.d. draw from a Plackett–Luce (PL) ranking model with $\exp(r)$ as the weight of the alternatives. Note that the marginal distribution induced by PL on any two alternatives follows BT. Create three pairwise preference datasets $\mathcal{D}_{r,12}$, $\mathcal{D}_{r,23}$, and $\mathcal{D}_{r,13}$ where $\mathcal{D}_{r,ij} = \{\boldsymbol{y}_i R \boldsymbol{y}_j \mid R \in P_r\}$. The social function $f$ applies $\mathcal{A}$ to every dataset to obtain a relation over $\mathcal{Y}_3$. By construction, $f$ is neutral and IIA.

By assumption, for every internal dataset $\mathcal{D}_{r,ij}$ we have $\Pr\left(\mathcal{A}(\mathcal{D}_{r,ij}) = R_{r,ij}\right) \geq 1 - \epsilon$, where $R_{r,ij}$ is the projection of $R_{\bar{r}}$ to only two alternatives $\boldsymbol{y}_i$ and $\boldsymbol{y}_j$. Using union bound, we have

$$\Pr\left(f(P_r) = R_{\bar{r}}\right) \geq 1 - 3\epsilon.$$

Similar to strategy 1, we can define a new reward function $r_\gamma$ with $\bar{r}_\gamma = \bar{r}$ such that the above holds for every $r_\gamma$ with $\gamma > 1$. Then, by increasing $\gamma$, the profile $P_{r_\gamma}$ approaches a uniformly distributed profile $P_{\mathrm{unif}}$. In this case, since $R_{\bar{r}}$ is an order relation, we have $\Pr\left(f \text{ is rational}\right) \geq 1 - 3\epsilon$. Then Theorem 1.3 of Friedgut et al. [44] implies that for some global constant $K$,

$$\Pr\left(f \text{ is dictatorship}\right) \geq 1 - 3K\epsilon.$$

On the other hand, we know that the order that $R_{\bar{r}}$ induces for different $r$ is not a dictatorship, so, we have

$$\Pr\left(f \text{ is dictatorship}\right) < 3\epsilon.$$

Putting these together, we obtain a lower bound on $\epsilon$:

$$\epsilon \geq \frac{1}{3(K+1)} > 0.$$

The rest of the proof is similar for the second and third strategies. We can use a boosting argument to show that from any weak pairwise learner, we can obtain an arbitrarily strong weak learner corresponding at the cost of collecting a larger dataset. We show this when for the weak learner $\epsilon < \frac{1}{2}$ but a weaker condition of choosing $R_{\bar{r}}$ better than chance level is sufficient for our argument. Consider a pairwise preference dataset $\mathcal{D}_r$ of size $m\, n_{\bar{r}}$. Partition $\mathcal{D}_r$ into $m$ equal-size datasets. Let $R_i$ be the output of $\mathcal{A}$ on the $i^{\mathrm{th}}$ dataset. Since samples in $\mathcal{D}_r$ are independently generated, $R_i$s err independently. Construct a meta-algorithm $\mathcal{A}_{\mathrm{maj}}$ that outputs the majority winner of $R_i$s. A standard Hoeffding bound implies

$$\Pr\left(\mathcal{A}_{\mathrm{maj}}(\mathcal{D}_r) \neq R_{\bar{r}}\right) \leq \exp\left(-2(1/2 - \epsilon)^2\, m\right).$$

A simple calculation then shows that for any arbitrarily small $\epsilon' > 0$, by choosing $m = O\left(\log(\frac{1}{\epsilon'})\left(\frac{1}{2} - \epsilon\right)^{-2}\right)$ the majority-winner algorithm agrees with $R_{\bar{r}}$ with probability of at least $1 - \epsilon'$. This contradicts the lower bound we established earlier and completes the proof.

**Proof Strategy 3.** The proof is by contradiction and is inspired by Procaccia et al. [41]. This proof requires at least four different user types. Suppose there are two equally represented user types $\mathcal{U} = \{A, B\}$ who follow BT. For some arbitrary response $\boldsymbol{y}_0 \in \mathcal{Y}$ and $0 < \tau < \frac{1}{3}$, consider the following reward function:

$$r_\tau(\boldsymbol{y}; u) = \begin{cases} 0, & \boldsymbol{y} \neq \boldsymbol{y}_0, \\ \sigma^{-1}(\frac{2}{3} + \tau) = \log \frac{\frac{2}{3} - \tau}{\frac{1}{3} + \tau}, & \boldsymbol{y} = \boldsymbol{y}_0, u = A, \\ \sigma^{-1}(\frac{2}{3} - \tau) = \log \frac{\frac{2}{3} + \tau}{\frac{1}{3} - \tau}, & \boldsymbol{y} = \boldsymbol{y}_0, u = B. \end{cases} \tag{21}$$

One can see $\Pr(\boldsymbol{y}_0 \succ \boldsymbol{y} \mid r_\tau) = \frac{2}{3}$ for every $\boldsymbol{y} \neq \boldsymbol{y}_0$. Therefore, $r_\tau$ induces the same pairwise preference distribution for every $\tau$. On the other hand, $\bar{r}_\tau(\boldsymbol{y}_0) := \mathbb{E}_u[r_\tau(\boldsymbol{y}_0; u)] = \log \frac{\frac{4}{9} - \tau^2}{\frac{1}{9} - \tau^2} > 0$ is an increasing function of $\tau$.

Consider two arbitrary $\tau_1$ and $\tau_2$ such that $0 < \tau_1 < \tau_2 < \frac{1}{3}$. Sample a pairwise preference dataset at follows. Draw a random $u$ and a random permutation $\rho$ over $\mathcal{Y}$. If $\rho$ is not identity, ask an annotator of type $u$ with reward $r_{\tau_1}(\cdot; u)$ to label this sample and permute the ranking with $\rho$. If $\rho$ is identity, ask an annotator of type $u$ with reward $r_{\tau_2}(\cdot; u)$ to label. This sampling is equivalent to sampling from $2 * |\mathcal{Y}|!$ different user types. By symmetry, this pairwise preference dataset is distributionally equivalent to a dataset $\mathcal{D}_{\text{unif}}$ with indifferent preferences. However, our construction implies that $\boldsymbol{y}_0$ has the highest expected reward and other alternatives have similar rewards:

$$\bar{r}(\boldsymbol{y}) = \frac{1}{|\mathcal{Y}|} \begin{cases} \bar{r}_{\tau_1}(\boldsymbol{y}), & \boldsymbol{y} \neq \boldsymbol{y}_0, \\ \bar{r}_{\tau_2}(\boldsymbol{y}), & \boldsymbol{y} = \boldsymbol{y}_0. \end{cases}$$

Denote the ranking based on $\bar{r}$ above by $R_0$.

Similar to the second strategy, suppose there exists an algorithm $\mathcal{A}$ that for any reward function $r$ and any preference dataset $\mathcal{D}_r$ with $|\mathcal{D}_r| \geq n_{\bar{r}}$, it outputs $R_{\bar{r}}$ with a probability of at least $1 - \epsilon$. Collect a preference dataset as explained above with at least $n_{\bar{r}}$ samples. Then, by assumption,

$$\Pr\left(\mathcal{A}(\mathcal{D}_{\text{unif}}) = R_0\right) \geq 1 - \epsilon.$$

Note that our choice of $\boldsymbol{y}_0$ could be any of the alternatives. Therefore, the above should be true when any of the alternatives has the highest expected reward. This implies a lower bound on $\epsilon$:

$$\epsilon \geq 1 - \frac{1}{|\mathcal{Y}|} > 0.$$

The rest of the proof is similar to the second strategy. $\qquad\square$

**Proposition 5.2.** *There exists no consistent M-estimator that without annotator information can estimate* $V(\boldsymbol{x}, \boldsymbol{y}_1, \boldsymbol{y}_2) := \operatorname{Var}_u\left[\Delta r^*(\boldsymbol{x}, \boldsymbol{y}_1, \boldsymbol{y}_2; u)\right]$.

*Proof of Proposition 5.2.* Consider a dataset $\mathcal{D}$ of context, candidate pairs, and preference represented as $(\boldsymbol{x}, \boldsymbol{y}_1, \boldsymbol{y}_2, o)$, where $o = \mathbb{1}\{\boldsymbol{y}_2 \succ \boldsymbol{y}_1\}$, and $\boldsymbol{y}_1, \boldsymbol{y}_2$ are independently drawn. Then consider an M-estimator

$$\underset{V}{\arg\min} \sum_{(\boldsymbol{x}, \boldsymbol{y}_l, \boldsymbol{y}_w) \in \mathcal{D}} \rho(\boldsymbol{x}, \boldsymbol{y}_l, \boldsymbol{y}_w; V) = \sum_{(\boldsymbol{x}, \boldsymbol{y}_1, \boldsymbol{y}_2, o) \in \mathcal{D}} o \cdot \rho(\boldsymbol{x}, \boldsymbol{y}_1, \boldsymbol{y}_2; V) + (1 - o) \cdot \rho(\boldsymbol{x}, \boldsymbol{y}_2, \boldsymbol{y}_1; V).$$

Under preference model of Eq. (2), for a reward $r$, we have $\mathbb{E}[o \mid \boldsymbol{x}, \boldsymbol{y}_1, \boldsymbol{y}_2; u] = \sigma\left(\Delta r(\boldsymbol{x}, \boldsymbol{y}_1, \boldsymbol{y}_2; u)\right)$. In the limit of a large dataset, the $M$-estimator solves

$$\underset{V}{\arg\min} \ \mathbb{E}_{\boldsymbol{x}, \boldsymbol{y}_1, \boldsymbol{y}_2, o, u}\left[o \cdot \rho(\boldsymbol{x}, \boldsymbol{y}_1, \boldsymbol{y}_2; V) + (1 - o) \cdot \rho(\boldsymbol{x}, \boldsymbol{y}_2, \boldsymbol{y}_1; V)\right]$$

$$= \mathbb{E}_{\boldsymbol{x}, \boldsymbol{y}_1, \boldsymbol{y}_2, u}\left[\sigma\left(\Delta r(\boldsymbol{x}, \boldsymbol{y}_1, \boldsymbol{y}_2; u)\right) \cdot \rho(\boldsymbol{x}, \boldsymbol{y}_1, \boldsymbol{y}_2; V) + \sigma\left(\Delta r(\boldsymbol{x}, \boldsymbol{y}_2, \boldsymbol{y}_1; u)\right) \cdot \rho(\boldsymbol{x}, \boldsymbol{y}_2, \boldsymbol{y}_1; V)\right] \quad (V \text{ has no } o)$$

$$= \mathbb{E}_{\boldsymbol{x}, \boldsymbol{y}_1, \boldsymbol{y}_2}\left[\mathbb{E}_u\left[\sigma\left(\Delta r(\boldsymbol{x}, \boldsymbol{y}_1, \boldsymbol{y}_2; u)\right)\right] \cdot \rho(\boldsymbol{x}, \boldsymbol{y}_1, \boldsymbol{y}_2; V) + \mathbb{E}_u\left[\sigma\left(\Delta r(\boldsymbol{x}, \boldsymbol{y}_2, \boldsymbol{y}_1; u)\right)\right] \cdot \rho(\boldsymbol{x}, \boldsymbol{y}_2, \boldsymbol{y}_1; V)\right] \quad (V \text{ has no } u)$$

$$= 2 \mathbb{E}_{\boldsymbol{x}, \boldsymbol{y}_1, \boldsymbol{y}_2}\left[\mathbb{E}_u\left[\sigma\left(\Delta r(\boldsymbol{x}, \boldsymbol{y}_1, \boldsymbol{y}_2; u)\right)\right] \cdot \rho(\boldsymbol{x}, \boldsymbol{y}_1, \boldsymbol{y}_2; V)\right]. \quad (\boldsymbol{y}_1, \boldsymbol{y}_2 \text{ are i.i.d})$$

Here, we used the fact that $V$ does not depend on $u$. We also relied on the assumption that $\boldsymbol{y}_1$ and $\boldsymbol{y}_2$ are identically and independently distributed. The above equation suggests that regardless of how $\rho$ is designed, $V(\boldsymbol{x}, \boldsymbol{y}_1, \boldsymbol{y}_2)$ can only depend on $u$'s distribution through $\mathbb{E}_u\left[\sigma\left(\Delta r(\boldsymbol{x}, \boldsymbol{y}_1, \boldsymbol{y}_2; u)\right)\right]$. Therefore, no consistent M-estimator can generally estimate $\operatorname{Var}_u\left[\Delta r(\boldsymbol{x}, \boldsymbol{y}_1, \boldsymbol{y}_2; u)\right]$ even with the availability of infinite preference data. $\qquad\square$

**Lemma 5.3.** *Using $J_1$ and $J_2$ as shorthands for $J(\boldsymbol{x}, \boldsymbol{y}_1, \boldsymbol{y}_2, \boldsymbol{x}, \boldsymbol{y}_1, \boldsymbol{y}_2)$ and $J(\boldsymbol{x}, \boldsymbol{y}_1, \boldsymbol{y}_2, \boldsymbol{x}, \boldsymbol{y}_2, \boldsymbol{y}_1)$, we can use the following to consistently estimate the variance term:*

$$V(\boldsymbol{y}_1, \boldsymbol{y}_2) = \frac{J_1 - (J_1 + J_2)^2}{\sigma'\left(\Delta \bar{r}^*(\boldsymbol{y}_1, \boldsymbol{y}_2)\right)^2}. \tag{16}$$

*Proof of Lemma 5.3.* First of all, $J$ can give us the likelihood itself:

$$J(\boldsymbol{x}, \boldsymbol{y}_1, \boldsymbol{y}_2, \boldsymbol{x}, \boldsymbol{y}_1, \boldsymbol{y}_2) + J(\boldsymbol{x}, \boldsymbol{y}_1, \boldsymbol{y}_2, \boldsymbol{x}, \boldsymbol{y}_2, \boldsymbol{y}_1)$$

$$= \mathbb{E}_u\left[\sigma\left(\Delta r^*(\boldsymbol{y}_1, \boldsymbol{y}_2; u)\right)^2 + \sigma\left(\Delta r^*(\boldsymbol{y}_1, \boldsymbol{y}_2; u)\right) \cdot \sigma\left(\Delta r^*(\boldsymbol{y}_2, \boldsymbol{y}_1; u)\right)\right]$$

$$= \mathbb{E}_u\left[\sigma\left(\Delta r^*(\boldsymbol{y}_1, \boldsymbol{y}_2; u)\right)\right].$$

Here, we used the property $\sigma\big(\Delta r^*(\boldsymbol{y}_2, \boldsymbol{y}_1; u)\big) = 1 - \sigma\big(\Delta r^*(\boldsymbol{y}_1, \boldsymbol{y}_2; u)\big)$. We also dropped $\boldsymbol{x}$ from the notation for simplicity. Since $J$ can give us both the first and second moments, we can use it to find $\mathrm{Var}_u\big[\sigma\big(\Delta r^*(\boldsymbol{x}, \boldsymbol{y}_1, \boldsymbol{y}_2; u)\big)\big]$ as follows:

$$
\begin{aligned}
&\mathrm{Var}_u\big[\sigma\big(\Delta r^*(\boldsymbol{x}, \boldsymbol{y}_1, \boldsymbol{y}_2; u)\big)\big] \\
&= \mathbb{E}_u\big[\sigma\big(\Delta r^*(\boldsymbol{x}, \boldsymbol{y}_1, \boldsymbol{y}_2; u)\big)^2\big] - \mathbb{E}_u\big[\sigma\big(\Delta r^*(\boldsymbol{x}, \boldsymbol{y}_1, \boldsymbol{y}_2; u)\big)\big]^2 \\
&= J(\boldsymbol{x}, \boldsymbol{y}_1, \boldsymbol{y}_2, \boldsymbol{x}, \boldsymbol{y}_1, \boldsymbol{y}_2) - \big(J(\boldsymbol{x}, \boldsymbol{y}_1, \boldsymbol{y}_2, \boldsymbol{x}, \boldsymbol{y}_1, \boldsymbol{y}_2) + J(\boldsymbol{x}, \boldsymbol{y}_1, \boldsymbol{y}_2, \boldsymbol{x}, \boldsymbol{y}_2, \boldsymbol{y}_1)\big)^2 .
\end{aligned}
$$

In the last piece of the proof, we connect $\mathrm{Var}_u\big[\sigma(\Delta r^*)\big]$ with $\mathrm{Var}_u\big[\Delta r^*\big]$. The Taylor expansion of $\sigma(\Delta r^*)$ around $\Delta \bar{r}^* := \mathbb{E}_u[\Delta r^*]$ gives

$$
\begin{aligned}
\mathrm{Var}_u\big[\sigma(\Delta r^*)\big] &= \mathrm{Var}_u\Big[\sigma(\Delta \bar{r}^*) + \sigma'(\Delta \bar{r}^*) \cdot (\Delta r^* - \Delta \bar{r}^*) + O\big((\Delta r^* - \Delta \bar{r}^*)^2\big)\Big] \\
&= \sigma'(\Delta \bar{r}^*)^2 \cdot \mathrm{Var}_u\big[\Delta r^*\big] + O\Big(\mathbb{E}_u\big[(\Delta r^* - \Delta \bar{r}^*)^3\big]\Big) .
\end{aligned}
$$

We can neglect the third-order term in calculations as first-order correction uses up to $O\Big(\mathbb{E}_u\big[(\Delta r^* - \Delta \bar{r}^*)^2\big]\Big)$ in its approximation. $\qquad\square$

**Proposition 6.1.** *Defining $l$ in Eq. (17) as follows results in a consistent estimation of the optimal policy when preferences follow the BT model:*

$$
l(\boldsymbol{y}_1, \boldsymbol{y}_2, \boldsymbol{o}; \pi) = \begin{cases} -\log \sigma\big(|\mathcal{U}| \cdot h(\boldsymbol{y}_1, \boldsymbol{y}_2; \pi)\big), & \boldsymbol{o} = \mathbf{1}, \\ -\log \sigma\big(|\mathcal{U}| \cdot h(\boldsymbol{y}_2, \boldsymbol{y}_1; \pi)\big), & \boldsymbol{o} = \mathbf{0}, \\ 0 & o.w. \end{cases}
$$

*Here, $h$ is the difference of $\pi$'s induced rewards (Eq. (15)), and $\mathbf{1}$ ($\mathbf{0}$) is the vector of all ones (zeros).*

*Proof of Proposition 6.1.* The proof follows similar steps as the derivation of DPO. First of all, conditioned on agreement, the likelihood of observing $\boldsymbol{y}_2 \succ \boldsymbol{y}_1$ under the BT model is

$$
\begin{aligned}
\Pr(\boldsymbol{y}_2 \succ \boldsymbol{y}_1 \mid r^*, \text{agreement}) &= \frac{\Pi_u \sigma\big(\Delta r^*(\boldsymbol{y}_1, \boldsymbol{y}_2; u)\big)}{\Pi_u \sigma\big(\Delta r^*(\boldsymbol{y}_1, \boldsymbol{y}_2; u)\big) + \Pi_u \sigma\big(\Delta r^*(\boldsymbol{y}_2, \boldsymbol{y}_1; u)\big)} \\
&= \frac{\exp\big(\sum_u r^*(\boldsymbol{y}_2; u)\big)}{\exp\big(\sum_u r^*(\boldsymbol{y}_1; u)\big) + \exp\big(\sum_u r^*(\boldsymbol{y}_2; u)\big)} \\
&= \sigma\Big(|\mathcal{U}| \cdot \mathbb{E}_u\big[\Delta r^*(\boldsymbol{y}_1, \boldsymbol{y}_2)\big]\Big) .
\end{aligned}
$$

On the other hand, Eq. (11) allows us to write $\mathbb{E}_u\big[\Delta r^*(\boldsymbol{y}_1, \boldsymbol{y}_2)\big]$ with difference $\pi$'s induced rewards, i.e., $h$:

$$
\Pr(\boldsymbol{y}_2 \succ \boldsymbol{y}_1 \mid \pi^*, \text{agreement}) = \sigma\big(h(\boldsymbol{y}_1, \boldsymbol{y}_2; \pi^*)\big) .
$$

We can define the likelihood in this way for every policy $\pi$. Then, the proposed loss function is equivalent to maximizing log-likelihood, which under mild conditions is a consistent estimator for $\pi^*$. $\qquad\square$

**Theorem 6.2.** *Suppose $l$ in Eq. (17) only depends on $(\boldsymbol{x}, \boldsymbol{y}_1, \boldsymbol{y}_2)$ through $\pi$ and $\pi_{\mathrm{ref}}$. If there are more than three types of user and the preferences follow BT, any loss that allows a consistent estimation of the optimal policy discards samples with disagreement, i.e., those with $\boldsymbol{o} \notin \{\mathbf{0}, \mathbf{1}\}$.*

*Proof of Theorem 6.2.* The proof involves three steps: First, the next lemma shows that any loss function $l$ in Eq. (17) with the desired consistency property can only depend on $\pi$ through the ratio $\frac{\pi(\boldsymbol{y}_2|\boldsymbol{x})}{\pi(\boldsymbol{y}_1|\boldsymbol{x})}$.

**Lemma G.1.** *Suppose $l$ in Eq. (17) only depends on $(\boldsymbol{x}, \boldsymbol{y}_1, \boldsymbol{y}_2)$ through $\pi$ and $\pi_{\mathrm{ref}}$. Then, for any $l$ that gives a consistent estimation of the optimal policy in Eq. (10), there exists an equivalent loss $\tilde{l}$ such that*

$$
l\big(\boldsymbol{x}, \boldsymbol{y}_1, \boldsymbol{y}_2, \boldsymbol{o}; \pi\big) = \tilde{l}\big(\boldsymbol{o}; h(\boldsymbol{x}, \boldsymbol{y}_1, \boldsymbol{y}_2; \pi)\big) ,
$$

*where $h$ is defined in Eq. (15).*

See proof on page 32.

In the second step, we further limit the search space of $\tilde{l}$ (as introduced by Lemma G.1) to those that meet certain first- and second-order conditions:

**Lemma G.2.** *Any loss $\tilde{l}$ as in Lemma G.1 that leads to a consistent estimation of the optimal policy meets*

$$\sum_{\boldsymbol{o}\in\{0,1\}^{\mathcal{U}}} \frac{\partial \tilde{l}}{\partial \theta}\big(\boldsymbol{o}; \theta^*(\boldsymbol{z})\big) \cdot \chi_{\boldsymbol{o}}(\boldsymbol{z}) = 0 \,,$$

$$\sum_{\boldsymbol{o}\in\{0,1\}^{\mathcal{U}}} \frac{\partial^2 \tilde{l}}{\partial \theta^2}\big(\boldsymbol{o}; \theta^*(\boldsymbol{z})\big) \cdot \chi_{\boldsymbol{o}}(\boldsymbol{z}) \geq 0 \,,$$

*for every $\boldsymbol{z} \in [0,1]^{\mathcal{U}}$. Here, we define*

$$\chi_{\boldsymbol{o}}(\boldsymbol{z}) := \prod_{u\in\mathcal{U}} z_u^{o_u}(1-z_u)^{1-o_u} \,,$$

*and*

$$\theta^*(\boldsymbol{z}) := \frac{1}{|\mathcal{U}|}\sum_{u\in\mathcal{U}} \sigma^{-1}(z_u) \,.$$

See proof on page 33.

Finally, we show that when preferences follow the BT model and there are more than three user types, all $\tilde{l}(\boldsymbol{o};\theta)$ terms corresponding to $\boldsymbol{o} \notin \boldsymbol{0}, \boldsymbol{1}$ do not depend on $\theta$. Therefore, these terms do not depend on $\pi$ and can be removed from the loss function, thereby completing the proof.

**Lemma G.3.** *If $|\mathcal{U}| > 3$, for any loss $\tilde{l}$ that meets the first-order condition of Lemma G.2, we have $\frac{\partial \tilde{l}}{\partial \theta}(\boldsymbol{o};\theta) = 0$ for every $\boldsymbol{o} \notin \{\boldsymbol{0}, \boldsymbol{1}\}$.*

See proof on page 33.

$\square$

**Proposition F.1.** *There exists a mixture of BTs that a single BT cannot represent.*

*Proof of Proposition F.1.* Suppose the pairwise comparison distribution over a set of alternatives $(\boldsymbol{y}_1, \boldsymbol{y}_2, \boldsymbol{y}_3, \dots)$ satisfies the Bradley-Terry (BT) model; i.e. $\Pr(\boldsymbol{y}_i \succ \boldsymbol{y}_j) = \sigma\big(r^*(\boldsymbol{y}_2) - r^*(\boldsymbol{y}_1)\big)$. Then:

$$\Pr(\boldsymbol{y}_1 \succ \boldsymbol{y}_2)\Pr(\boldsymbol{y}_2 \succ \boldsymbol{y}_3)\Pr(\boldsymbol{y}_3 \succ \boldsymbol{y}_1) = \frac{\prod_{i=1}^3 \exp(r^*(\boldsymbol{y}_i))}{\prod_{i=1}^3 \big(\exp(r^*(\boldsymbol{y}_i)) + \exp(r^*(\boldsymbol{y}_{(i+1)\bmod 3+1}))\big)}$$

$$= \Pr(\boldsymbol{y}_1 \succ \boldsymbol{y}_3)\Pr(\boldsymbol{y}_3 \succ \boldsymbol{y}_2)\Pr(\boldsymbol{y}_2 \succ \boldsymbol{y}_1) \,.$$

Now, consider two BT models corresponding to $u_1$ and $u_2$, with a uniform mixture over them. For the mixture:

$$\Pr(\boldsymbol{y}_i \succ \boldsymbol{y}_j) = \frac{\Pr(\boldsymbol{y}_i \succ \boldsymbol{y}_j \mid u_1) + \Pr(\boldsymbol{y}_i \succ \boldsymbol{y}_j \mid u_2)}{2} \,.$$

The probability of cyclic preferences in one direction is given by

$$\Pr(\boldsymbol{y}_1 \succ \boldsymbol{y}_2)\Pr(\boldsymbol{y}_2 \succ \boldsymbol{y}_3)\Pr(\boldsymbol{y}_3 \succ \boldsymbol{y}_1) = \frac{\sum_{s\in\{1,2\}^3}\prod_{i=1}^3 \Pr(\boldsymbol{y}_i \succ \boldsymbol{y}_{(i+1)\bmod 3+1} \mid u_{s_i})}{8} \,,$$

which is not necessarily equal to the probability of the cyclic preferences in the reverse direction:

$$\Pr(\boldsymbol{y}_1 \succ \boldsymbol{y}_3)\Pr(\boldsymbol{y}_3 \succ \boldsymbol{y}_2)\Pr(\boldsymbol{y}_2 \succ \boldsymbol{y}_1) = \frac{\sum_{s\in\{1,2\}^3}\prod_{i=1}^3 \Pr(\boldsymbol{y}_{(i+1)\bmod 3+1} \succ \boldsymbol{y}_i \mid u_{s_i})}{8} \,.$$

To verify this, consider specific examples such as $\Pr(\boldsymbol{y}_i \succ \boldsymbol{y}_j \mid u_k) = \frac{\exp(r_k^i)}{\exp(r_k^i)+\exp(r_k^j)}$ with $r_1 = (1,2,3)$ and $r_2 = (1,2,4)$. More generally, the BT assumption implies that, for a fixed

reward $r^*$, the likelihood of a set of pairwise comparisons $\{(\boldsymbol{y}_{p,1} > \boldsymbol{y}_{p,2})\}_{p \in [P]}$ is proportional to $\prod_i \exp(r^*(\boldsymbol{y}_i))^{|\{p \in [P] \mid \boldsymbol{y}_{p,1}=i\}|}$ and depends only on the number of times each option is preferred in the comparisons. However, as demonstrated above, this property does not hold for a mixture of BT models. $\square$

**Proposition F.3.** *Defining $l$ in Eq. (17) as follows results in a consistent estimation of the optimal policy when preferences follow the BT model:*

$$l(\boldsymbol{y}_1, \boldsymbol{y}_2, \boldsymbol{o}; \pi) = \begin{cases} -\sigma\big(h(\boldsymbol{y}_1, \boldsymbol{y}_2; \pi)\big) - I\big(\sigma\big(h(\boldsymbol{y}_1, \boldsymbol{y}_2; \pi)\big)\big), & \boldsymbol{o} = \mathbf{1}\,, \\ -\sigma\big(h(\boldsymbol{y}_2, \boldsymbol{y}_1; \pi)\big) - I\big(\sigma\big(h(\boldsymbol{y}_2, \boldsymbol{y}_1; \pi)\big)\big), & \boldsymbol{o} = \mathbf{0}\,, \\ 0 & o.w. \end{cases}$$

*Here, we define $I(\theta) := \int_1^\theta \left(\frac{1}{\theta'} - 1\right)^{|\mathcal{U}|} \mathrm{d}\theta'$, and $h$ is the difference of $\pi$'s induced rewards (Eq. (15)).*

*Proof of Proposition F.3.* Recall $\Pr(o_u = 1 \mid \boldsymbol{x}, \boldsymbol{y}_1, \boldsymbol{y}_2) = \sigma\big(\Delta r^*(\boldsymbol{x}, \boldsymbol{y}_1, \boldsymbol{y}_2; u)\big)$. We use $z_u(\boldsymbol{x}, \boldsymbol{y}_1, \boldsymbol{y}_2)$ as a shorthand for this quantity and will drop the dependence on $(\boldsymbol{x}, \boldsymbol{y}_1, \boldsymbol{y}_2)$ whenever it is clear from the context. We also use $s$ as a shorthand for $\sigma(h)$. In the limit of a very large dataset, the proposed loss approaches

$$\mathcal{L}(s) = -\mathbb{E}_{\boldsymbol{x}, \boldsymbol{y}_1, \boldsymbol{y}_2}\Big[\big(\prod_{u \in \mathcal{U}} z_u\big)\big(s + I(s)\big) + \big(\prod_{u \in \mathcal{U}}(1 - z_u)\big)\big(1 - s + I(1 - s)\big)\Big]\,.$$

Note that we wrote $\mathcal{L}$ as a function of $s$ instead of $\pi$ since $s$ is the only place that $\pi$ appears. We first show that $\mathcal{L}(s)$ has a unique global minimizer. To show an $s$ is a global minimizer of $\mathcal{L}$, it suffices to show that $s$ minimizes the term inside expectation for every $(\boldsymbol{x}, \boldsymbol{y}_1, \boldsymbol{y}_2)$. Such a minimizer meets the first-order condition:

$$\big(\prod_{u \in \mathcal{U}} z_u\big)\Big(1 + (\frac{1}{s} - 1)^{|\mathcal{U}|}\Big) + \big(\prod_{u \in \mathcal{U}}(1 - z_u)\big)\Big(-1 - (\frac{1}{1-s} - 1)^{|\mathcal{U}|}\Big) = 0\,.$$

Here, we used $\frac{\mathrm{d}I}{\mathrm{d}\theta} = (\frac{1}{\theta} - 1)^{|\mathcal{U}|}$. Define $w := (\frac{1-s}{s})^{|\mathcal{U}|}$. Then, the above condition reduces to a quadratic equation in terms of $w$:

$$1 + w - \big(\prod_{u \in \mathcal{U}}(\frac{1}{z_u} - 1)\big)(1 + w^{-1}) = (1 + w^{-1})\Big[w - \prod_{u \in \mathcal{U}}(\frac{1}{z_u} - 1)\Big] = 0\,.$$

Solving for $w$, we obtain

$$s^* = \frac{1}{1 + \big(\prod_{u \in \mathcal{U}}(\frac{1}{z_u} - 1)\big)^{\frac{1}{|\mathcal{U}|}}}\,.$$

For the BT model, a direct calculation then shows

$$s^*(\boldsymbol{x}, \boldsymbol{y}_1, \boldsymbol{y}_2) = \sigma\Big(\frac{1}{|\mathcal{U}|} \sum_{u \in \mathcal{U}} \Delta r(\boldsymbol{x}, \boldsymbol{y}_1, \boldsymbol{y}_2; u)\Big)\,. \tag{22}$$

In fact, $s^*$ is the only global minimizer of $\mathcal{L}(s)$. This is because $\mathcal{L}(s)$ is convex in $s$:

$$\frac{\mathrm{d}^2\mathcal{L}}{\mathrm{d}s^2} = \mathbb{E}_{\boldsymbol{x}, \boldsymbol{y}_1, \boldsymbol{y}_2}\Big[\big(\prod_{u \in \mathcal{U}} z_u\big) \cdot \frac{|\mathcal{U}|}{s^2}(\frac{1}{s} - 1)^{|\mathcal{U}|-1} + \big(\prod_{u \in \mathcal{U}}(1 - z_u)\big) \cdot \frac{|\mathcal{U}|}{(1-s)^2}(\frac{1}{1-s} - 1)^{|\mathcal{U}|-1}\Big] \geq 0\,.$$

Finally, one can verify that the policy that results in $s^*$ (Eq. (22)) is the optimal policy $\pi^*$. This completes the proof that the proposed loss is a consistent loss for $\pi^*$. $\square$

**Lemma G.1.** *Suppose $l$ in Eq. (17) only depends on $(\boldsymbol{x}, \boldsymbol{y}_1, \boldsymbol{y}_2)$ through $\pi$ and $\pi_{\mathrm{ref}}$. Then, for any $l$ that gives a consistent estimation of the optimal policy in Eq. (10), there exists an equivalent loss $\tilde{l}$ such that*

$$l\big(\boldsymbol{x}, \boldsymbol{y}_1, \boldsymbol{y}_2, \boldsymbol{o}; \pi\big) = \tilde{l}\big(\boldsymbol{o}; h(\boldsymbol{x}, \boldsymbol{y}_1, \boldsymbol{y}_2; \pi)\big)\,,$$

*where $h$ is defined in Eq. (15).*

*Proof of Lemma G.1.* Since $l$ only depends on $(\boldsymbol{x}, \boldsymbol{y}_1, \boldsymbol{y}_2)$ through $\pi$ and $\pi_{\text{ref}}$, we overload the notation and use $l\big(\boldsymbol{o}; \pi(\boldsymbol{y}_1 \mid \boldsymbol{x}), \pi(\boldsymbol{y}_2 \mid \boldsymbol{x})\big)$ to denote the loss from $(\boldsymbol{x}, \boldsymbol{y}_1, \boldsymbol{y}_2, \boldsymbol{o})$. In the limit of many data points, $\mathcal{L}(\mathcal{D}; \pi)$ converges to

$$\mathcal{L}(\pi) = \mathbb{E}_{\boldsymbol{x}, \boldsymbol{y}_1, \boldsymbol{y}_2, \boldsymbol{o}}\big[l\big(\boldsymbol{o}; \pi(\boldsymbol{y}_1 \mid \boldsymbol{x}), \pi(\boldsymbol{y}_2 \mid \boldsymbol{x})\big)\big] \, .$$

Using $\Pr(o_u = 1 \mid \boldsymbol{x}, \boldsymbol{y}_1, \boldsymbol{y}_2) = \sigma\big(\Delta r^*(\boldsymbol{x}, \boldsymbol{y}_1, \boldsymbol{y}_2; u)\big)$, the tower rule implies

$$\mathcal{L}(\pi) = \mathbb{E}_{\boldsymbol{x}, \boldsymbol{y}_1, \boldsymbol{y}_2}\Big[\mathbb{E}_{\boldsymbol{o}}\big[l\big(\boldsymbol{o}; \pi(\boldsymbol{y}_1 \mid \boldsymbol{x}), \pi(\boldsymbol{y}_2 \mid \boldsymbol{x})\big) \mid \boldsymbol{x}, \boldsymbol{y}_1, \boldsymbol{y}_2\big]\Big]$$

$$= \mathbb{E}_{\boldsymbol{x}, \boldsymbol{y}_1, \boldsymbol{y}_2}\Big[\sum_{\boldsymbol{o} \in \{0,1\}^{\mathcal{U}}} l\big(\boldsymbol{o}; \pi(\boldsymbol{y}_1 \mid \boldsymbol{x}), \pi(\boldsymbol{y}_2 \mid \boldsymbol{x})\big) \cdot \prod_{u \in \mathcal{U}} \sigma\big(\Delta r^*(\boldsymbol{x}, \boldsymbol{y}_1, \boldsymbol{y}_2; u)\big)^{o_u} \big(1 - \sigma\big(\Delta r^*(\boldsymbol{x}, \boldsymbol{y}_1, \boldsymbol{y}_2; u)\big)\big)^{1 - o_u}\Big] \, .$$

For notational simplicity, let's define

$$z_u(\boldsymbol{x}, \boldsymbol{y}_1, \boldsymbol{y}_2) := \sigma\big(\Delta r^*(\boldsymbol{x}, \boldsymbol{y}_1, \boldsymbol{y}_2; u)\big) \, ,$$

$$\chi_{\boldsymbol{o}}(\boldsymbol{z}) := \prod_{u \in \mathcal{U}} z_u^{o_u} (1 - z_u)^{1 - o_u} \, .$$

Note that $\chi_{\boldsymbol{o}}$ implicitly depends on $(\boldsymbol{x}, \boldsymbol{y}_1, \boldsymbol{y}_2)$ through $\boldsymbol{z}$, which we drop from the notation when it is clear from the context. Using this notation, we can rewrite the $\mathcal{L}(\pi)$'s expansion as follows:

$$\mathcal{L}(\pi) = \mathbb{E}_{\boldsymbol{x}, \boldsymbol{y}_1, \boldsymbol{y}_2}\Big[\sum_{\boldsymbol{o} \in \{0,1\}^{\mathcal{U}}} l\big(\boldsymbol{o}; \pi(\boldsymbol{y}_1 \mid \boldsymbol{x}), \pi(\boldsymbol{y}_2 \mid \boldsymbol{x})\big) \cdot \chi_{\boldsymbol{o}}\big(\boldsymbol{z}(\boldsymbol{x}, \boldsymbol{y}_1, \boldsymbol{y}_2)\big)\Big] \, . \tag{23}$$

Overloading notation, we can always equivalently represent $\big(\pi(\boldsymbol{y}_1 \mid \boldsymbol{x}), \pi(\boldsymbol{y}_2 \mid \boldsymbol{x})\big)$ as $\big(\pi(\boldsymbol{y}_1, \boldsymbol{y}_2 \mid \boldsymbol{x}), \pi(\boldsymbol{y}_2 \mid \boldsymbol{y}_1, \boldsymbol{y}_2, \boldsymbol{x})\big)$, that is, with the probability that either of the two responses is chosen and the probability that the second one is preferred. Therefore, there exists a loss $l'$ such that

$$l\big(\boldsymbol{o}; \pi(\boldsymbol{y}_1 \mid \boldsymbol{x}), \pi(\boldsymbol{y}_2 \mid \boldsymbol{x})\big) = l'\big(\boldsymbol{o}; \pi(\{\boldsymbol{y}_1, \boldsymbol{y}_2\} \mid \boldsymbol{x}), \pi(\boldsymbol{y}_2 \mid \{\boldsymbol{y}_1, \boldsymbol{y}_2\}, \boldsymbol{x})\big) \, .$$

If $\pi$ is optimal, $\pi(\boldsymbol{y}_2 \mid \boldsymbol{y}_1, \boldsymbol{y}_2, \boldsymbol{x})$ should also be optimal. Since $\pi(\boldsymbol{y}_2 \mid \boldsymbol{y}_1, \boldsymbol{y}_2, \boldsymbol{x})$ appears in only one term of the expectation in Eq. (23), we can conclude that

$$\arg\min_{\theta'} \sum_{\boldsymbol{o} \in \{0,1\}^{\mathcal{U}}} l'\big(\boldsymbol{o}; \pi^*(\{\boldsymbol{y}_1, \boldsymbol{y}_2\} \mid \boldsymbol{x}), \theta'\big) \cdot \chi_{\boldsymbol{o}}\big(\boldsymbol{z}(\boldsymbol{x}, \boldsymbol{y}_1, \boldsymbol{y}_2)\big)$$

is the optimal $\pi(\boldsymbol{y}_2 \mid \{\boldsymbol{y}_1, \boldsymbol{y}_2\}, \boldsymbol{x})$ for every optimal $\pi(\{\boldsymbol{y}_1, \boldsymbol{y}_2\} \mid \boldsymbol{x})$. On the other hand, a property of the optimal policy $\pi^*$ is that

$$\pi^*(\boldsymbol{y}_2 \mid \{\boldsymbol{y}_1, \boldsymbol{y}_2\}, \boldsymbol{x}) = \frac{\pi^*(\boldsymbol{y}_2 \mid \boldsymbol{x})}{\pi^*(\boldsymbol{y}_1 \mid \boldsymbol{x})} = \frac{\pi_{\text{ref}}(\boldsymbol{y}_2 \mid \boldsymbol{x})}{\pi_{\text{ref}}(\boldsymbol{y}_1 \mid \boldsymbol{x})} \cdot \exp\Big(\frac{1}{\beta}\mathbb{E}_u\big[\Delta r^*(\boldsymbol{x}, \boldsymbol{y}_1, \boldsymbol{y}_2; u)\big]\Big) \, .$$

Therefore, for every optimal policy $\pi^*(\{\boldsymbol{y}_1, \boldsymbol{y}_2\} \mid \boldsymbol{x})$, we have

$$\frac{\pi_{\text{ref}}(\boldsymbol{y}_2 \mid \boldsymbol{x})}{\pi_{\text{ref}}(\boldsymbol{y}_1 \mid \boldsymbol{x})} \cdot \exp\Big(\frac{1}{\beta}\mathbb{E}_u\big[\Delta r^*(\boldsymbol{x}, \boldsymbol{y}_1, \boldsymbol{y}_2; u)\big]\Big) =$$

$$\arg\min_{\theta'} \sum_{\boldsymbol{o} \in \{0,1\}^{\mathcal{U}}} l'\big(\boldsymbol{o}; \pi^*(\{\boldsymbol{y}_1, \boldsymbol{y}_2\} \mid \boldsymbol{x}), \theta'\big) \cdot \chi_{\boldsymbol{o}}\big(\boldsymbol{z}(\boldsymbol{x}, \boldsymbol{y}_1, \boldsymbol{y}_2)\big) \, . \tag{24}$$

Recall from the optimal policy (Eq. (10)) that we can modify the reward function for responses other than $(\boldsymbol{x}, \boldsymbol{y}_1, \boldsymbol{y}_2)$ while keeping $\Delta r^*(\boldsymbol{x}, \boldsymbol{y}_1, \boldsymbol{y}_2; u)$ constant. This allows arbitrary changes to $\pi^*(\{\boldsymbol{y}_1, \boldsymbol{y}_2\} \mid \boldsymbol{x})$ without altering the rest of Eq. (24). So, we can argue that $l'$ does not depend on $\pi(\{\boldsymbol{y}_1, \boldsymbol{y}_2\} \mid \boldsymbol{x})$ and we drop it from $l'$ notation. Define a new loss based on $l'$:

$$\tilde{l}(\boldsymbol{o}; \theta) := l'\Big(\boldsymbol{o}; \frac{\pi_{\text{ref}}(\boldsymbol{y}_2 \mid \boldsymbol{x})}{\pi_{\text{ref}}(\boldsymbol{y}_1 \mid \boldsymbol{x})} \cdot \exp\big(\frac{1}{\beta}\theta\big)\Big) \, .$$

Note that $\tilde{l}$ implicitly depends on $(\boldsymbol{x}, \boldsymbol{y}_1, \boldsymbol{y}_2)$ through $\pi_{\text{ref}}$ which we dropped from notation. Using $\tilde{l}$, we can write the original loss $l$ as

$$l\big(\boldsymbol{o}; \pi(\boldsymbol{y}_1 \mid \boldsymbol{x}), \pi(\boldsymbol{y}_2 \mid \boldsymbol{x})\big) = l'\big(\boldsymbol{o}; \pi(\boldsymbol{y}_2 \mid \{\boldsymbol{y}_1, \boldsymbol{y}_2\}, \boldsymbol{x})\big)$$

$$= l'\Big(\boldsymbol{o}; \frac{\pi(\boldsymbol{y}_2 \mid \boldsymbol{x})}{\pi(\boldsymbol{y}_1 \mid \boldsymbol{x})}\Big)$$

$$= \tilde{l}\Big(\boldsymbol{o}; \beta \log \frac{\pi(\boldsymbol{y}_2 \mid \boldsymbol{x})}{\pi(\boldsymbol{y}_1 \mid \boldsymbol{x})} - \beta \log \frac{\pi_{\text{ref}}(\boldsymbol{y}_2 \mid \boldsymbol{x})}{\pi_{\text{ref}}(\boldsymbol{y}_1 \mid \boldsymbol{x})}\Big)$$

$$= \tilde{l}\big(\boldsymbol{o}; h(\boldsymbol{x}, \boldsymbol{y}_1, \boldsymbol{y}_2; \pi)\big) \, .$$

This completes the proof. □

**Lemma G.2.** *Any loss $\tilde{l}$ as in Lemma G.1 that leads to a consistent estimation of the optimal policy meets*

$$\sum_{\boldsymbol{o}\in\{0,1\}^{\mathcal{U}}} \frac{\partial\tilde{l}}{\partial\theta}\big(\boldsymbol{o};\theta^*(\boldsymbol{z})\big)\cdot\chi_{\boldsymbol{o}}(\boldsymbol{z}) = 0\,,$$

$$\sum_{\boldsymbol{o}\in\{0,1\}^{\mathcal{U}}} \frac{\partial^2\tilde{l}}{\partial\theta^2}\big(\boldsymbol{o};\theta^*(\boldsymbol{z})\big)\cdot\chi_{\boldsymbol{o}}(\boldsymbol{z}) \geq 0\,,$$

*for every $\boldsymbol{z}\in[0,1]^{\mathcal{U}}$. Here, we define*

$$\chi_{\boldsymbol{o}}(\boldsymbol{z}) := \prod_{u\in\mathcal{U}} z_u^{o_u}(1-z_u)^{1-o_u}\,,$$

*and*

$$\theta^*(\boldsymbol{z}) := \frac{1}{|\mathcal{U}|}\sum_{u\in\mathcal{U}}\sigma^{-1}(z_u)\,.$$

*Proof of Lemma G.2.* We will refer to the proof of Lemma G.1 in this proof. Using $\tilde{l}$ in place of $l'$ in Eq. (24), since $\exp(\cdot)$ is monotone increasing, we have

$$\mathbb{E}_u\big[\Delta r^*(\boldsymbol{x},\boldsymbol{y}_1,\boldsymbol{y}_2;u)\big] = \arg\min_\theta \sum_{\boldsymbol{o}\in\{0,1\}^{\mathcal{U}}} \tilde{l}(\boldsymbol{o};\theta)\cdot\chi_{\boldsymbol{o}}(\boldsymbol{z})\,.$$

On the other hand, using the fact that user types in $\mathcal{U}$ are equiprobable, we can write

$$\mathbb{E}_u\big[\Delta r^*(\boldsymbol{x},\boldsymbol{y}_1,\boldsymbol{y}_2;u)\big] = \frac{1}{|\mathcal{U}|}\sum_{u\in\mathcal{U}}\sigma^{-1}(z_u)\,.$$

Putting these together, it is necessary to have

$$\frac{1}{|\mathcal{U}|}\sum_{u\in\mathcal{U}}\sigma^{-1}(z_u) = \arg\min_\theta \sum_{\boldsymbol{o}\in\{0,1\}^{\mathcal{U}}} \tilde{l}(\boldsymbol{o};\theta)\cdot\chi_{\boldsymbol{o}}(\boldsymbol{z})$$

for every $\boldsymbol{z}\in[0,1]^{\mathcal{U}}$. The rest of the proof is straightforward. □

**Lemma G.3.** *If $|\mathcal{U}| > 3$, for any loss $\tilde{l}$ that meets the first-order condition of Lemma G.2, we have $\frac{\partial\tilde{l}}{\partial\theta}(\boldsymbol{o};\theta) = 0$ for every $\boldsymbol{o}\notin\{\boldsymbol{0},\boldsymbol{1}\}$.*

*Proof of Lemma G.3.* First of all, for the BT model, a direct calculation shows

$$\theta^*(\boldsymbol{z}) := \frac{1}{|\mathcal{U}|}\sum_{u\in\mathcal{U}}\sigma^{-1}(z_u) = \frac{1}{|\mathcal{U}|}\log\prod_{u\in\mathcal{U}}\Big(\frac{z_u}{1-z_u}\Big) = \frac{1}{|\mathcal{U}|}\log\Big(\frac{\chi_{\boldsymbol{1}}}{\chi_{\boldsymbol{0}}}\Big)\,.$$

Since $\theta^*(\boldsymbol{z})$ depends on $\boldsymbol{z}$ only through $\frac{\chi_{\boldsymbol{0}}}{\chi_{\boldsymbol{1}}}$, we denote $\frac{\partial\tilde{l}}{\partial\theta}(\boldsymbol{o};\theta^*)$ by $g\big(\boldsymbol{o};\frac{\chi_{\boldsymbol{0}}}{\chi_{\boldsymbol{1}}}\big)$. The proof has two steps: In the first step, we relate $g\big(\boldsymbol{o};\frac{\chi_{\boldsymbol{0}}}{\chi_{\boldsymbol{1}}}\big)$ to $g\big(\boldsymbol{o}^{\oplus u'};\frac{\chi_{\boldsymbol{0}}}{\chi_{\boldsymbol{1}}}\big)$, where we define

$$\boldsymbol{o}^{\oplus u'} := \begin{cases} o_u\,, & u\neq u'\,, \\ 1-o_u\,, & u=u'\,. \end{cases}$$

Using this connection, in the second step, we will show that $g\big(\boldsymbol{o};\frac{\chi_{\boldsymbol{0}}}{\chi_{\boldsymbol{1}}}\big) = 0$ for any $\boldsymbol{o}\in\{\boldsymbol{0},\boldsymbol{1}\}$ when $|\mathcal{U}|\geq 4$.

**Step 1.** Consider any $\boldsymbol{o}\notin\{\boldsymbol{0},\boldsymbol{1}\}$. When $|\mathcal{U}|\geq 3$, there exists $u'\in\mathcal{U}$ such that $\boldsymbol{o}^{\oplus u'}\notin\{\boldsymbol{0},\boldsymbol{1}\}$. For such $\boldsymbol{o}$ and $u'$, we define two non-empty sets

$$\mathcal{S}^1 := \{u \mid u\in\mathcal{U}, u\neq u', o_u=1\}\,,$$
$$\mathcal{S}^0 := \{u \mid u\in\mathcal{U}, u\neq u', o_u=0\}\,.$$

We set $z$ such that

$$\prod_{u \in \mathcal{S}^1} z_u = \prod_{u \in \mathcal{S}^0} (1 - z_u) \,,$$
$$z_u \to 1^- \,, \ \forall u \in \mathcal{S}^1 \,,$$
$$z_u \to 0^+ \,, \ \forall u \in \mathcal{S}^0 \,.$$

(25)

For this choice of $z$, asymptotically, we have

$$\frac{\chi_{\mathbf{0}}}{\chi_{\mathbf{1}}} = \frac{1 - z_{u'}}{z_{u'}} \,,$$
$$\chi_{\mathbf{o}} = z_{u'}^{o_{u'}} (1 - z_{u'})^{1 - o_{u'}} \,,$$
$$\chi_{\mathbf{o} \oplus u'} = z_{u'}^{1 - o_{u'}} (1 - z_{u'})^{o_{u'}} \,,$$
$$\chi_{\mathbf{o}'} = 0 \,, \ \forall \mathbf{o}' \notin \{\mathbf{o}, \mathbf{o}^{\oplus u'}\} \,.$$

Using the above, we can simplify the first-order condition in Lemma G.2 as

$$g\big(\mathbf{o}; \frac{1 - z_{u'}}{z_{u'}}\big) \cdot z_{u'}^{o_{u'}} (1 - z_{u'})^{1 - o_{u'}} + g\big(\mathbf{o}^{\oplus u'}; \frac{1 - z_{u'}}{z_{u'}}\big) \cdot z_{u'}^{1 - o_{u'}} (1 - z_{u'})^{o_{u'}} = 0 \,.$$

This condition should be held for every $z_{u'} \in [0, 1]$. Therefore, we can conclude

$$g(\mathbf{o}^{\oplus u'}; \alpha) = -g(\mathbf{o}; \alpha) \cdot \alpha^{1 - 2o_{u'}} \,,$$

(26)

for every $\alpha \in \mathbb{R}$. This completes the first part of the proof.

**Step 2.** When $\mathbf{o} \notin \{\mathbf{0}, \mathbf{1}\}$ and $|\mathcal{U}| \geq 4$, there exist distinct user types $u'$ and $u''$ such that none of $\mathbf{o}^{\oplus u'}$, $\mathbf{o}^{\oplus u''}$, and $\mathbf{o}^{\oplus(u', u'')}$ are in $\{\mathbf{0}, \mathbf{1}\}$. Here, we used $\mathbf{o}^{\oplus(u', u'')}$ as a shorthand for $(\mathbf{o}^{\oplus u'})^{\oplus u''}$. For such $\mathbf{o}$, $u'$, and $u''$, we define two non-empty sets

$$\mathcal{S}^1 := \{u \mid u \in \mathcal{U} \setminus \{u', u''\}, o_u = 1\} \,,$$
$$\mathcal{S}^0 := \{u \mid u \in \mathcal{U} \setminus \{u', u''\}, o_u = 0\} \,.$$

We set $z$ according to Eq. (25). Then, asymptotically,

$$\chi_{\mathbf{o}'} = 0 \,, \ \forall \mathbf{o}' \notin \{\mathbf{o}, \mathbf{o}^{\oplus u'}, \mathbf{o}^{\oplus u''}, \mathbf{o}^{\oplus(u', u'')}\} \,.$$

Using the above, we can simplify the first-order condition in Lemma G.2 as

$$g\big(\mathbf{o}; \frac{\chi_{\mathbf{0}}}{\chi_{\mathbf{1}}}\big) \cdot \chi_{\mathbf{o}} + g\big(\mathbf{o}^{\oplus u'}; \frac{\chi_{\mathbf{0}}}{\chi_{\mathbf{1}}}\big) \cdot \chi_{\mathbf{o} \oplus u'} + g\big(\mathbf{o}^{\oplus u''}; \frac{\chi_{\mathbf{0}}}{\chi_{\mathbf{1}}}\big) \cdot \chi_{\mathbf{o} \oplus u''} + g\big(\mathbf{o}^{\oplus(u', u'')}; \frac{\chi_{\mathbf{0}}}{\chi_{\mathbf{1}}}\big) \cdot \chi_{\mathbf{o} \oplus(u', u'')} = 0 \,.$$

(27)

Because of the symmetry of this equation, we can assume without loss of generality that $o_{u'} = 0$ and $o_{u''} = 1$. Therefore, asymptotically, we have

$$\frac{\chi_{\mathbf{0}}}{\chi_{\mathbf{1}}} = \big(\frac{1 - z_{u'}}{z_{u'}}\big)\big(\frac{1 - z_{u''}}{z_{u''}}\big) \,,$$
$$\chi_{\mathbf{o}} = (1 - z_{u'}) \cdot z_{u''} \,,$$
$$\chi_{\mathbf{o} \oplus u'} = z_{u'} \cdot z_{u''} \,,$$
$$\chi_{\mathbf{o} \oplus u''} = (1 - z_{u'}) \cdot (1 - z_{u''}) \,,$$
$$\chi_{\mathbf{o} \oplus(u', u'')} = z_{u'} \cdot (1 - z_{u''}) \,.$$

Eq. (26) also implies

$$g(\mathbf{o}^{\oplus u'}; \alpha) = -g(\mathbf{o}; \alpha) \cdot \alpha \,,$$
$$g(\mathbf{o}^{\oplus u''}; \alpha) = -g(\mathbf{o}; \alpha) \cdot \alpha^{-1} \,,$$
$$g(\mathbf{o}^{\oplus(u', u'')}; \alpha) = g(\mathbf{o}; \alpha) \,.$$

Plugging these into Eq. (27) and simplifying equations, we obtain

$$g\Big(\mathbf{o}; \big(\frac{1 - z_{u'}}{z_{u'}}\big)\big(\frac{1 - z_{u''}}{z_{u''}}\big)\Big) \cdot (2z_{u'} - 1)(2z_{u''} - 1) = 0 \,.$$

This equation should be held for every $z_{u'}$ and $z_{u''}$. By appropriately setting $z_{u'}$ and $z_{u''}$, we can conclude that $g(\mathbf{o}; \alpha)$ should be zero for every $\alpha$. This completes this proof. $\qquad\square$

