# OpenReview forum: "Direct Alignment with Heterogeneous Preferences"
_NeurIPS.cc/2025/Conference — NeurIPS 2025 poster_

### Official Review · Reviewer_wgAn · 2025-06-27

**Clarity:** 3
**Significance:** 4
**Originality:** 3
**Rating:** 5
**Confidence:** 3

**Summary:**

This paper investigates the challenge of learning an optimal policy from preference data when preferences are heterogeneous across multiple user types. The authors show that standard DPO methods, which assume homogeneous preferences, fail to capture this heterogeneity, leading to suboptimal policies. To address this, the authors theoretically prove that learning a consistent policy without annotator information is impossible and propose methods to overcome this limitation by leveraging minimal annotator information, such as paired preferences from the same user. They introduce a first-order corrected DPO algorithm that incorporates variance estimation to improve policy estimation. Furthermore, they develop a consistent loss function for scenarios with rich annotator data, enabling direct alignment with the optimal policy but at the cost of sample efficiency. Finally, the paper conducts synthetic and semi-synthetic experiments, showing that accounting for user heterogeneity yields policies that better align with average user rewards compared to existing methods.

**Questions:**

1. The proposed first-order variance correction relies on training a joint likelihood model to estimate the variance term (Eq.16). Could the authors provide a detailed analysis of the computational overhead incurred by this correction, particularly in large-scale settings?
2. The consistent loss function assumes access to the annotator identity for each sample and assumes all user types label each point. In practice, this is rarely the case. How does the method perform when some annotator identities are missing or when annotations are noisy or incomplete?
3. The comparisons focus primarily on standard DPO and its corrected variants. However, related works like EM-DPO and MODPO also address heterogeneous preferences.

**Ethical Concerns:**

["NO or VERY MINOR ethics concerns only"]

**Final Justification:**

After reading the author's rebuttal, I have decided to increase my original score.

**Limitations:**

yes

**Paper Formatting Concerns:**

No major formatting issues noticed.

**Quality:**

3

**Strengths And Weaknesses:**

Strengths

1. The paper explores and investigates the critical yet largely underexplored challenge of addressing heterogeneity in preference data within alignment and preference-based learning, aiming to better align language models with human objectives and values.
2. The paper proposes a first-order correction to DPO and develops consistent loss functions under different levels of annotator information.
3. The synthetic and semi-synthetic experiments seem well-designed to empirically validate theoretical claims, showing concrete improvements over baseline methods.

Weakness:
1. Some concepts on personalization and multi-user reward aggregation have appeared in prior work (e.g., EM-DPO, MODPO). However, this paper should set itself apart by providing stronger formal guarantees.
2. Limited generality due to reliance on DPO framework and BT model assumptions, which may not capture the complexity of real-world preference learning scenarios.
3. Heavy dependence on rich annotator information leads to impractical data requirements and poor sample efficiency
The proposed consistent loss function requires detailed annotator identity information for every sample and assumes all user types label each data point. This results in exponential data demands as the number of user types grows, which is unrealistic in many deployment scenarios where such fine-grained labeling is costly or infeasible.
4. The practicality and computational cost of the first-order variance correction have not been sufficiently analyzed. While the paper introduces a variance estimator via a joint likelihood model for first-order correction of DPO, the computational overhead and stability of this estimator in large-scale or real-world training remain unclear.
5. The experiments do not test robustness under more realistic conditions, such as noisy annotations and missing user types per sample. We would like to see that.

---

> ### Author Rebuttal · Authors · 2025-07-31
>
> We thank the reviewer for their thoughtful review and valuable comments. We fully agree that addressing heterogeneity in preferences is a critical yet largely underexplored challenge. We hope our strong theoretical contributions help advance understanding in this important area. We also appreciate that the reviewer found our experiments well-designed. Below, we address the specific points and questions raised.
>
> > Comparisons focus primarily on standard DPO and its corrected variants. However, related works like EM-DPO and MODPO also address heterogeneous preferences. … this paper should set itself apart by providing stronger formal guarantees.
>
> Several works have addressed alignment in the presence of heterogeneous preferences and we discuss them extensively in Appendix B. EM-DPO is the closest work to our setting in the sense that it aims to learn a single policy serving the whole population. However, it defines maximum regret among the user types as its objective for the final policy, which is inherently different from the more natural expected reward objective, and relies on a heuristic expectation maximization using stochastic gradient descent which comes with high computational complexity. Methods such as MODPO rely on other structures of preference data to capture heterogeneous preferences and personalized methods such as P-RLHF learn a separate policy for each user type, none of them making up a comparable baseline for our method.
>
> That being said, minimizing our consistent loss is theoretically optimal for our proposed objective under the rich data setting. Our aim in the experiments is to illustrate how much improvement is left on the table when heterogeneity of preferences is ignored. For this purpose, DPO serves as a natural benchmark due to its popularity and its homogeneity assumption.
>
> > Limited generality due to reliance on DPO framework and BT model assumptions.
>
> We appreciate the reviewer’s observation. Our main impossibility result (Thm. 6.2) is established under the BT preference model. However, Prop. 3.1 (which shows only affine aggregations lead to a well-defined problem), Prop. 5.1 (impossibility of learning an optimal policy with no annotator information), Prop. 5.2 (impossibility of M-estimation with no annotator info), and Lemma 5.3 (consistent variance estimation) all hold in full generality for an arbitrary preference model as defined in Eq. 2.
>
> That being said, BT is by far the most widely used preference model for post-training and alignment of language models. For example, OpenAI’s InstructGPT (Ouyang et al.’22), Google’s Gemma and Gemini models (Team et al.’24), all use BT for alignment, since it consistently outperforms alternative approaches (Liu et al.’24). Consequently it's also the most studied preference model which we adopt for our impossiblity result.
>
> We also emphasize that our main impossibility result applies to any direct alignment method—not just DPO or the DPO family. The only part of the paper which is specifically about DPO is Sec. 4, where we analyze the consequences of the homogeneity assumption in DPO as a flagship direct alignment method (even these results extend to RLHF by substituting the policy with its induced reward, but framing them under DPO simplifies the exposition).
>
> > Exponential data demands as the number of user types grows.
>
> Our goal in Sec. 6 is to find out what is and isn’t possible given rich annotator information. Specifically, we show that consistent learning of the optimal policy is possible and propose corresponding loss functions (Prop. 6.1 and Prop. C.3). However, as the reviewer points out, these loss functions become sample inefficient when there are many user types. Our main theoretical result, Thm. 6.2, shows that this inefficiency is not specific to our losses but is a fundamental limitation of any direct alignment method. Therefore, as the reviewer correctly notes, our impossibility results make direct alignment undesirable as the number of user types increases. As we discuss, one can then either forgo consistency and pursue approximate methods—such as the first-order correction to DPO—or resort to indirect methods that learn reward functions before policy learning.
>
> > Computational overhead of the first-order variance correction.
>
> Generally, optimizing the joint likelihood $J$ to obtain $V$ is roughly as challenging as training a policy or reward model, though the exact difficulty depends on the chosen architecture. Compared to a policy operating on $[x, y]$, $J$ takes as input $[x, y_1, y_2, x', y'_1, y'_2]$, potentially making its input context four times longer. However, since $J$ is a binary predictor, the task may be simpler than token-level prediction, depending on the model design.
>
> Despite the added complexity, learning $V$ is a one-time cost and does not scale with the number of user types. As a result, learning $V$ and direct alignment is still more efficient than training separate reward models for each user type followed by RL. In our paper, we evaluate the first-order correction of DPO using estimated $V$ in synthetic settings. While this offers some initial insight, we agree with the reviewer that real-world experiments and stability analyses can be next steps. These remain open directions, as our current work focuses on the theoretical challenges posed by heterogeneous preferences, where the first-order correction illustrates one way in which relaxing consistency can lead to efficient learning that was otherwise impossible due to Thm. 6.2. We do not present it as a definitive solution, but as a step toward deeper exploration of this trade-off and hope to encourage further work into practical and scalable approaches for handling diverse preferences.
>
> > Experiments for robustness under noisy annotations.
>
> We agree that evaluating the robustness of the consistent loss function under noisy annotations is an interesting direction. Given the short rebuttal period, we have simulated this in our synthetic experiment in Sec. 7.1. To model noisy preferences, for a noise level $n \le 1$, we sample the preference of a type-$u$ user from a random user type with probability $n$. We then compare the learned policies under different noise levels with the optimal policy (OPT), reporting total variation (TV), Kendall’s $\tau$, and Spearman’s correlation. The latter two measure ordinal similarity between policies.
>
> As the table below shows, the TV between the noisy $\pi_{\rm consistent}$ and OPT drops below that of DPO and OPT only at high noise levels ($n \ge 0.7$). Even at those levels, $\pi_{\rm consistent}$ remains ordinally closer to OPT than DPO’s policy. These early observations suggest that the benefits of consistent losses may be robust to annotation noise. However, we emphasize that these findings are limited to synthetic settings and may not generalize broadly, but surely motivate further theoretical and empirical work on the robustness aspect.
>
> Importantly, this robustness does not mitigate the sample inefficiency of direct alignment methods: Our key impossibility result still discourages their use in settings with many user types. We thank the reviewer for this question and will include this discussion in the updated paper.
>
> | Metric             | DPO   | $\pi_{\rm consistent}$ (noise=0.0) | noise=0.1 | noise=0.2 | noise=0.3 | noise=0.4 | noise=0.5 | noise=0.6 | noise=0.7 |
> |--------------------|-------|------------------------------|----------|----------|----------|----------|----------|----------|----------|
> | TV                 | 0.089 | 0.023                        | 0.044    | 0.066    | 0.073    | 0.081    | 0.084    | 0.088    | 0.090    |
> | Kendall's $\tau$   | 0.526 | 0.933                        | 0.888    | 0.757    | 0.710    | 0.668    | 0.649    | 0.618    | 0.563    |
> | Spearman's $r$     | 0.676 | 0.989                        | 0.976    | 0.912    | 0.863    | 0.814    | 0.811    | 0.762    | 0.727    |

---

### Official Review · Reviewer_Fxtn · 2025-07-01

**Clarity:** 3
**Significance:** 3
**Originality:** 3
**Rating:** 5
**Confidence:** 3

**Summary:**

This paper studies and formalizes the tradeoff between ambiguity that arises from preference feedback learning and sample efficiency. The paper then proposes a first order correction to DPO, and show that minimal information can improve upon standard DPO formulation.

**Questions:**

See the point on real world experiments in the weaknesses section.

**Ethical Concerns:**

["NO or VERY MINOR ethics concerns only"]

**Final Justification:**

I read all the responses and maintain my opinion post rebuttal.

**Limitations:**

yes

**Quality:**

3

**Strengths And Weaknesses:**

Strengths:

a. theoretical as well as empirical results that support the claims

b. the paper concretely formalizes this tradeoff of maximizing average reward vs the average of different rewards and how that conflicts with sample efficiency.
In particular, the authors illustrate how DPO makes a crucial assumption that the average of the preferences is roughly the same as the preference of the average.

c. I also appreciated the insight that this assumption relies on variance in annotator labels, i.e their disagreement, leading to the proposed first order correction.

Weaknesses:

Lack of real world experiments that show this variance/annotator disagreement in practice and the improvement with the correction to DPO formulation.

---

> ### Author Rebuttal · Authors · 2025-07-31
>
> We thank the reviewer for their thoughtful feedback and positive assessment. We are glad the reviewer found both our theoretical and empirical results compelling, and appreciated our formalization of the tradeoff arising in maximizing average of different rewards. We are especially pleased that the reviewer highlighted our analysis of DPO’s key assumption and its connection to annotator disagreement, which motivates our proposed first-order correction.
>
> > Real-world experiments that show this variance/annotator disagreement in practice
>
> We agree with the reviewer that our main contributions are theoretical, highlighting the tensions inherent in using any direct alignment method under heterogeneous preferences. To motivate the existence of variance or annotator disagreement, we primarily draw on prior studies and provide motivating examples based on real-world PEW surveys (which were noted as helpful by Reviewer cLLj). We agree that annotator disagreement and variance are important topics and, in light of our results, believe they warrant greater attention. We hope our work encourages further empirical research in this direction.

---

> > ### Comment · Reviewer_Fxtn · 2025-08-06
> >
> > I read the above response and other reviews/responses and maintain my score post rebuttal.

---

### Official Review · Reviewer_yY38 · 2025-07-01

**Clarity:** 3
**Significance:** 3
**Originality:** 3
**Rating:** 5
**Confidence:** 4

**Summary:**

This paper investigates the alignment of a single AI policy with diverse human preferences, a setting it calls "heterogeneous preferences". The authors argue that the common approach of assuming a single reward function is flawed and propose that the goal should be to optimize the average reward across different "user types". They demonstrate that standard methods like Direct Preference Optimization (DPO) fail to achieve this; instead, DPO aligns with the Normalized Borda Count, which can promote mediocre outcomes and is sensitive to the data sampling distribution. The paper proves that learning the optimal policy is impossible without annotator information but introduces solutions for different information levels: a "first-order corrected DPO" using minimal data (paired preferences) and a consistent loss function when full annotator feedback is available. Ultimately, the research reveals a fundamental trade-off, showing that any direct alignment method that is consistent in this rich data setting is inherently sample-inefficient, as it must discard data where users disagree.

**Questions:**

Could the authors provide the approximation error estimation in Equation (14) and using $V$ to approximate the variance. Since the authors mentioned the Sensitivity to Preference Dataset Distribution in Line 167.

**Ethical Concerns:**

["NO or VERY MINOR ethics concerns only"]

**Final Justification:**

The reviewers have resolved my question. I maintain my score to be accept.

**Limitations:**

Yes.

**Paper Formatting Concerns:**

No.

**Quality:**

4

**Strengths And Weaknesses:**

### Strength
- The paper demonstrates that standard methods like Direct Preference Optimization (DPO) fail to optimize for this average reward. Instead, DPO implicitly optimizes for the "Normalized Borda Count" (NBC), a voting rule that can lead to undesirable outcomes, such as promoting mediocre options and being sensitive to preference dataset distribution.
- The paper tackles a problem of paramount importance: how to create a single AI system that serves a pluralistic society with conflicting preferences. Its primary contribution is a fundamental negative result (Theorem 6.2) that establishes a tension between consistency and sample efficiency for direct alignment methods. This is a highly significant finding, as it delineates the boundaries of what is possible for an entire class of popular alignment techniques.
### Weaknesses
- Key impossibility results are proven under the assumption of the Bradley-Terry (BT) preference model, which may limit the generality of the conclusions.

---

> ### Author Rebuttal · Authors · 2025-07-31
>
> We thank the reviewer for their helpful and insightful comments. We’re glad that they found our problem of paramount importance and our main impossibility result a highly significant finding.
>
> > Key impossibility results are proven under the assumption of the BT preference model, which may limit the generality of the conclusions.
>
> We appreciate the reviewer’s observation. Our main impossibility result (Thm. 6.2) is established under the BT preference model. However, Prop. 3.1 (which shows only affine aggregations lead to a well-defined problem), Prop. 5.1 (impossibility of learning an optimal policy with no annotator information), Prop. 5.2 (impossibility of M-estimation with no annotator info), and Lemma 5.3 (consistent variance estimation) all hold in full generality for an arbitrary preference model as defined in Eq. 2.
>
> That being said, BT is by far the most widely used preference model for post-training and alignment of language models. For example, OpenAI’s InstructGPT [1], Google’s Gemma and Gemini models [2], all use BT for alignment, since it consistently outperforms alternative approaches [3]. Consequently, it's the most popular model to stufy human preferences too.
>
> > Could the authors provide the approximation error estimation in Eq. 14 and using $V$ to approximate the variance. Since the authors mentioned the Sensitivity to Preference Dataset Distribution in Line 167.
>
> We thank the reviewer for the clarifying questions. Eq. 14 omits $O(\mathbb{E}_u[(\Delta r^* - \Delta \bar{r})^3])$ and higher-order terms, as analyzed in the proof of Lemma 5.3 (line 824). The variance estimator $V$ in Lemma 5.3 is not an approximation but a consistent estimator of the variance. Its performance, however, depends on optimization quality and finite-sample effects, which lie beyond the scope of our population-level analysis. Regarding sensitivity to the preference data distribution as defined in line 167, our $V$ estimator is not sensitive in this sense, since it consistently estimates $\text{Var}_u[\Delta r^*]$, which is itself independent of the data distribution. Similarly, the consistent losses discussed in Sec. 6 are also not sensitive in this sense, as their optima do not depend on the data distribution. We thank the reviewer for these questions and will clarify them in the revision.
>
> [1] Ouyang et al. (2022). Training language models to follow instructions with human feedback. Advances in Neural Information Processing Systems
>
> [2] Team et al. (2024). Gemma: Open models based on Gemini research and technology
>
> [3] Liu et al. (2024). Skywork-reward: Bag of tricks for reward modeling in LLMs.

---

> > ### Comment · Reviewer_yY38 · 2025-08-07
> >
> > Thanks for your reply. I will maintain my score.

---

### Official Review · Reviewer_cLLj · 2025-07-03

**Clarity:** 3
**Significance:** 3
**Originality:** 3
**Rating:** 4
**Confidence:** 3

**Summary:**

This paper studies heterogeneous user preferences and address alignment problems in this setting. Theoretically, it proves that any single policy compatible with multiple user types must reduce to an affine combination of rewards. It further shows that standard Direct Preference Optimization (DPO) is ordinally consistent with the Normalized Borda Count (NBC) rather than with that average reward . To mitigate the mismatch, the authors propose a first-order corrected DPO for minimal annotator information, and a consistent loss that learns the optimal policy when every data point is labelled by all user types. Experiments on Pew surveys, synthetic toy tasks, and semi-synthetic fine-tuning of Llama-3-8B and Qwen-2.5-7B show gains in rewards when heterogeneity is modeled.

**Questions:**

1) In practice, what guidance can you give for grouping user types? Do authors have ablation study on the sensitivity to the grouping choice?
2) If users are mis-clustered or have no types, how do the first-order and consistent objectives degrade?
3) Can this framework bring benefits if take the preference heterogeneity into account during inference?

**Ethical Concerns:**

["NO or VERY MINOR ethics concerns only"]

**Final Justification:**

After reading the rebuttal, I decide to maintain my score.

**Limitations:**

yes

**Quality:**

3

**Strengths And Weaknesses:**

Strengths:
1) The paper is well written and structured.
2) The authors identify the problem of heterogeneous user preferences, providing solid theoretical characterizations for the optimal policy, and the 'objective' DPO is trying to optimize. These results provide valuable insights in modeling preference data and designing leaning algorithms.
3) The PEW Research Center surveys demonstrate that NBC can conflict with average reward in practice, which justify the theoretical findings.


Weaknesses:
The LLM experiments rely on length-based semi-synthetic rewards. It is hard to judge if the methodology can bring practical benefits.

---

> ### Author Rebuttal · Authors · 2025-07-31
>
> We appreciate the reviewer's insightful feedback and thoughtful comments. We are pleased that they found our paper well-written and well-structured, offering valuable insights into modeling heterogeneous preferences and developing algorithms. Below, we address the reviewer’s questions.
>
> > If users are mis-clustered or have no types, how do the first-order and consistent objectives degrade?
>
> * If users have no type, i.e., homogeneous preferences, the first-order correction and consistent objectives (e.g., Prop. 6.1 and Prop. C.3) reduce to standard DPO.
> * If users have different types but no annotator information is available, then neither the optimal policy nor any approximation of it can be learned consistently (Prop. 5.1 and 5.2).
> * If users have different types and minimal annotation information (i.e., paired preferences), we propose a consistent first-order correction of DPO (Lemma 5.3). This method does not require explicit knowledge of user types or any clustering—it remains valid with many types and without identifying them.
> * If users have different types and we know the type and collect a rich dataset where each sample is labeled by a representative of each type, we propose consistent loss functions for directly learning the optimal policy. However, our main impossibility result shows that even with correctly specified types, neither these nor any direct alignment methods are sample efficient for many user types. When user types are misclustered, i.e., incorrectly specified, we believe learning in this setting with the consistent objective only becomes harder. We have provided a new synthetic experiment showing this and the robustness of the consistent objective in our response to Reviewer wgAn, but we do not have any formal guarantee at the moment. In our response to the next question we also discuss how our results can help mitigate such misclusterings.
>
> > What guidance can you give for grouping user types?
>
> While our study focuses on the consequences of the existence of user types, we agree that discovering them is also important and valuable. Our theory and algorithms can inform user type discovery as follows. Let $\mathcal{U}$ denote the set of true user types, and let $u'$ be a proposed group that may include users from multiple true types. A direct extension of Eq. 14 suggests that the inaccuracy from modeling $u'$ as a single type in DPO-style algorithms is proportional to
> $$V(x, y_1, y_2 \mid u') := \text{Var}_i[\Delta r^*(x, y_1, y_2; u(i)) \mid i \in u'],$$
>
> where $u(i)$ denotes the true type of user $i$. Given $V$, one can define $l(u') := \sum_{(x, y_1, y_2) \in \mathcal{D}} V(x, y_1, y_2 \mid u')$ as a measure of the homogeneity of $u'$, and accept $u’$ as a single user type if $l(u') < \epsilon$ for a practical choice of $\epsilon$. Note that this will only give the criterion for an acceptable user type, but how to find such types is an algorithmic question (for example, agglomerative or divisive clustering might work here).
>
> The bottleneck in this clustering approach is estimating $V$. As Prop. 5.2 shows, no M-estimator can consistently estimate $V$ without annotator information. On the positive side, our algorithm in Lemma 5.3 can estimate $V$ consistently using minimal annotator information, namely, paired preferences (Sec. 5.3). However, repeatedly estimating $V$ in an algorithm can be computationally intensive.
>
> In summary, our theory provides a criterion to test when a group of users can be treated as a single type. This requires estimating a variance term, which we show is feasible using a joint likelihood maximization with minimal information. While our tools are useful for user type discovery, we acknowledge the computational challenges and view the development of more efficient algorithms as an important direction for future work. We thank the reviewer for raising this point and will include our response in the revision.
>
> > Can this framework bring benefits if take the preference heterogeneity into account during inference?
>
> This is an intriguing question. Although we are not entirely sure how alignment can occur during inference in our context, inference-time (re-)alignment of LLMs is a lively research area. Recent studies have investigated methods such as best-of-N sampling (Huang et al.’25), activation steering (Wang et al.’24), transferring logit differences from smaller aligned models (Mitchell et al.’23), diffusion-style alignment (Chen et al.’25), integrated value guidance (Liu et al.’24), and reward-guided generation for test-time (re-)alignment (Xu et al.’24). Combining these techniques with our heterogeneity-aware methodology will be an exciting direction for future work!

---

### Decision · Program_Chairs · 2025-09-17

**Decision:**

Accept (poster)

**Comment:**

This paper investigates alignment under heterogeneous human preferences, formalizing the problem using user types and analyzing the limitations of the common homogeneity assumption. The authors show that aligning a single policy to multiple user types reduces to optimizing an affine combination of rewards, but standard Direct Preference Optimization (DPO) instead aligns with the Normalized Borda Count, which can diverge from the average reward. To address this mismatch, the paper introduces (i) a first-order corrected DPO that improves performance with minimal annotator information, and (ii) consistent loss functions that learn the optimal policy when full annotator information is available. Both theoretical analysis and experiments on Pew survey data, synthetic tasks, and semi-synthetic LLM fine-tuning (Llama-3-8B, Qwen-2.5-7B) illustrate the importance of modeling heterogeneity and highlight a fundamental tension between consistency and sample efficiency in direct alignment methods.

The reviewers agreed that the problem is timely and important, with valuable insights into preference-based alignment. Strengths noted include the clear identification of the mismatch between DPO and average reward, the introduction of corrected objectives under different information settings, and a fundamental impossibility result. Concerns focused on the limitations of the theoretical framework, particularly its reliance on the Bradley–Terry preference model (yY38, wgAn), the strong data requirements of the consistent loss objectives (wgAn), and the lack of large-scale or real-world empirical validation beyond semi-synthetic settings (cLLj, Fxtn). The authors adequately clarified how their results generalize beyond BTL, and supplemented the rebuttal with new synthetic experiments.

Overall, I recommend acceptance. The work provides a rigorous and timely contribution to the study of pluralistic alignment, clarifying both what is possible and what is fundamentally limited when aligning to heterogeneous preferences. Despite some practical concerns regarding scalability and empirical breadth, the theoretical insights and accompanying experiments significantly advance understanding in this area. Please incorporate all the suggestions of the reviewers in the final version as promised.